# Beyond Simple Graphs: Neural Multi-Objective Routing on Multigraphs

**Filip Rydin**[*1]**, Attila Lischka**[1]**, Jiaming Wu**[1]**,**
**Morteza Haghir Chehreghani**[1,2] **& Balázs Kulcsár**[1]

[1]Chalmers University of Technology, [2]University of Gothenburg

## Abstract

Learning-based methods for routing have gained significant attention in recent years, both in single-objective and multi-objective contexts. Yet, existing methods are unsuitable for routing on multigraphs, which feature multiple edges with distinct attributes between node pairs, despite their strong relevance in real-world scenarios. In this paper, we propose two graph neural network-based methods to address multi-objective routing on multigraphs. Our first approach operates directly on the multigraph by autoregressively selecting edges until a tour is completed. The second model, which is more scalable, first simplifies the multigraph via a learned pruning strategy and then performs autoregressive routing on the resulting simple graph. We evaluate both models empirically, across a wide range of problems and graph distributions, and demonstrate their competitive performance compared to strong heuristics and neural baselines[**].

## 1 Introduction

The field of neural combinatorial optimization has grown significantly in recent years and vehicle routing problems in particular have attracted much attention (Zhou et al., 2024a). While early works focused on the Traveling Salesman Problem (TSP) (Vinyals et al., 2015; Bello et al., 2017), new learning-based methods for vehicle routing can solve a wide range of problems efficiently and effectively, often surpassing classical state-of-the-art heuristics (Zhou et al., 2024b; Drakulic et al., 2025). Yet, existing methods have one limitation in common: they assume problems are defined on simple graphs. However, multigraph formulations, featuring several edges between each node pair, become relevant as soon as there are competing edges that cannot be chosen between a priori. Such situations typically occur when edges have more than one feature of interest, such as both travel time and distance.

In spite of the high practical relevance of multigraph formulations (Lai et al., 2016; Ben Ticha et al., 2017), current learning-based methods are not equipped to handle them due to two main reasons. Firstly, many state-of-the-art neural solvers rely on transformers to encode the problem instance. While these work well in the Euclidean setting (Kool et al., 2018) and with some modifications on asymmetric, directed graphs (Kwon et al., 2021), they are ill-suited for encoding multigraph structures. Secondly and more importantly, planning routes in multigraphs requires both selecting the node order and which edges to traverse, making most current decoding strategies unsuitable.

In this work, we aim to bridge the gap between learning-based methods for routing and accurate network representations given by multigraphs. We focus on the Multi-Objective (MO) setting, as several competing objectives naturally translates to several competing edges between each node pair. Nevertheless, our methods are general and can easily be extended to single-objective settings. Concretely, our main contributions are:

- We design models to handle multigraph inputs. To our knowledge, we present the first neural solvers specifically designed for such structures. In Appendix B, we argue further why current methods are insufficient, even when combined with graph transformations.

---

[*]Correspondence to: `filipry@chalmers.se`

[**]Our implementation is available at `https://github.com/filiprydin/GMS`.

- Our models are also capable of handling a variety of MO routing problems on asymmetric graphs. In contrast, current neural multi-objective methods are either problem-specific or designed exclusively for the Euclidean setting, where edge costs are symmetric and determined only by the node coordinates. The non-Euclidean, asymmetric setting is generally viewed as more practically relevant (Boyacı et al., 2021). In our results, we show that naive extensions of many single-objective neural methods for asymmetric problems fail to perform well, illustrating the difficulty of the multi-objective non-Euclidean setting.
- We propose two separate models based on Graph Neural Networks (GNNs): One edge-based working directly on the multigraph and one dual-task model that first prunes the multigraph into a simple graph and then selects routes. The former is simple but slow while the latter is more complex but faster.
- Experimentally, we show that both our approaches achieve competitive results for several routing problems on asymmetric graphs and multigraphs, including variants of the multi-objective TSP and multi-objective capacitated vehicle routing problem.

## 2 RELATED WORK

### 2.1 LEARNING FOR ROUTING

Motivated by the success of end-to-end learning in fields such as image classification and language modeling, neural solutions for routing have gained increasing attention during the last years. Broadly, methods can be classified by how they obtain tours. Two prominent alternatives are *autoregressive construction*, based on iterative building of tours one node at a time, and *non-autoregressive construction*, that outputs heatmaps in a one-shot fashion for down-stream decoding with some search method. Examples from the first category include Vinyals et al. (2015); Kool et al. (2018); Kwon et al. (2020). Examples from the second category are Joshi et al. (2019); Fu et al. (2021); Sun & Yang (2023). For a more comprehensive discussion of these methods, we refer the reader to Appendix A, which presents additional related work.

The literature on MO routing is more sparse than in the single-objective case. A key contribution was made by Lin et al. (2022), who proposed an autoregressive construction approach using a single neural model to learn the Pareto set of solutions. Regarding asymmetric problems, the only existing learning-based methods we are aware of are those introduced by Santiyuda et al. (2024) and Zhou et al. (2025). However, both these approaches are tailored to specific problems and have been evaluated within narrowly defined domains. In contrast, our method is capable of addressing a broad class of routing problems and is evaluated across various problems and settings.

### 2.2 ROUTING ON MULTIGRAPHS

The multigraph representation has gained increasing attention in the operations research literature on Vehicle Routing Problems (VRPs) in recent years. Garaix et al. (2010) performed the first study in this regard and later works include those by Lai et al. (2016); Ben Ticha et al. (2017; 2019) and Tikani et al. (2021). A consistent finding is that the multigraph representation leads to considerably improved solutions compared to planning on simple graphs. For example, in the VRP with time windows, Ben Ticha et al. (2017) reported cost reductions of up to $10.5\%$ on real-world instances, while Lai et al. (2016) found average savings of around $5\%$ for a heterogeneous VRP. However, these benefits typically come at the cost of increased problem complexity. Learning-based methods can thus fill a clear need for new solvers in this area, as they have proven to be efficient, effective and broadly applicable. To the best of our knowledge, this work presents the first learning-based approach to address combinatorial routing problems such as the TSP and VRPs on multigraphs. In contrast, current studies either use traditional heuristics or exact solvers.

## 3 PROBLEM FORMULATION

We define a general multi-objective routing problem as

$$\min_{\pi \in \Pi} f(\pi), \tag{1}$$

where $f : \Pi \rightarrow \mathbb{R}^m$ with $m > 1$. Here, $\pi \in \Pi$ represents a feasible route in the multigraph $G$, encoded as a sequence of edges, and $f(\pi)$ is the associated cost vector.

The solution to an MO routing problem is a set of objective values known as the Pareto Front (PF) with associated routes in the Pareto Set (PS). Formally, we define the PS as

$$\text{PS} := \{\pi \in \Pi \,|\, \nexists \pi' \in \Pi : f(\pi') \prec f(\pi)\}, \tag{2}$$

where $f(\pi') \prec f(\pi)$ denotes that $f(\pi')$ dominates $f(\pi)$, meaning

$$\begin{aligned} f_i(\pi') \leq f_i(\pi) &\quad \forall i = 1, \ldots, m, \\ f_i(\pi') < f_i(\pi) &\text{ for at least one } i. \end{aligned} \tag{3}$$

Correspondingly, the Pareto Front (PF) is defined as the image of the Pareto Set under $f$, that is,

$$\text{PF} := \{f(\pi) \in \mathbb{R}^m \,|\, \pi \in \text{PS}\}. \tag{4}$$

Intuitively, the PS represents solutions for which no strictly better alternative exists.

To obtain this set, we utilize a typical method: decomposition through scalarization. The idea is to translate the MO problem into several subproblems that can be solved separately. Let $\lambda \in \mathbb{R}^m$ such that $\lambda_i \geq 0 \ \forall i$ and $\sum_i \lambda_i = 1$ be a vector that controls the preference among the objectives. Then, under *Linear Scalarization* (LS), the single-objective cost is $f_\lambda(\pi) = \sum_i \lambda_i f_i(\pi)$ and the corresponding subproblem is $\min_\pi f_\lambda(\pi)$. Unfortunately, it is well-known that there may be solutions in the PS that do not solve an LS subproblem for any preference $\lambda$ (Ehrgott, 2005). The *Chebyshev scalarization* has more favorable properties. Let $z_i^* < \min_\pi f_i(\pi)$ be an ideal value for objective $i$. Then, the Chebyshev scalarized cost is

$$f_\lambda(\pi) = \max_i \{\lambda_i |f_i(\pi) - z_i^*|\}. \tag{5}$$

Given a solution in the PS, there always exists a preference $\lambda$ and corresponding Chebyshev subproblem for which the solution is optimal (Choo & Atkins, 1983).

In our context of multigraphs, it is interesting to consider how problems simplify under scalarization. We highlight an important result for the Multi-Objective TSP (MOTSP), in which the total cost of a tour is the sum of the vectorial costs of the traversed edges. The proof of the following proposition is omitted due to its simplicity. Given a linear scalarization defined by a preference vector $\lambda$, define the reduced graph $G(\lambda)$ by removing any edge whose scalarized cost is strictly worse than that of a parallel edge.

**Proposition 1.** *Let $f_\lambda(\pi)$ denote the linearly scalarized cost and let $\Pi(\lambda) \subset \Pi$ be the set of feasible tours on the pruned graph $G(\lambda)$. Then, the optimal value of the scalarized subproblem is preserved:*

$$\min_{\pi \in \Pi} f_\lambda(\pi) = \min_{\pi \in \Pi(\lambda)} f_\lambda(\pi). \tag{6}$$

This illustrates that a multigraph representation is sometimes easy to handle using pruning. However, under Chebyshev scalarization or for more complex problems (e.g., with edge attributes that contribute nonlinearly to the cost), similar results might not be available. As such, we propose to learn which parallel edge is optimal while simultaneously learning to construct the optimal route for each preference $\lambda$. Moreover, by explicitly taking into account the multigraph representation when designing our models, we obtain better results and faster inference than if we pre-prune the multigraphs, even for the MOTSP.

## 4 GNN-BASED MULTIGRAPH SOLVER

In the following, we propose two approaches based on GNNs for solving MO multigraph routing problems. We call our method GNN-based Multigraph Solver (GMS) and propose an edge-based variant constructing routes autoregressively directly on the multigraph as well as a variant with two decoders that first prunes and then constructs routes.

Both variants require an encoder that i) can handle input data structured as multigraphs and ii) works with and outputs edge embeddings. While there are multiple GNN architectures that fit this description, we utilize the Graph Edge Attention Network (GREAT) of Lischka et al. (2025), since

it has been applied in an autoregressive framework and shown competitive performance for non-Euclidean single-objective routing problems.

Specifically, we utilize the Node-Based (NB) version of GREAT. We refer the reader to Lischka et al. (2025) for details. Here, it suffices to say that the core component of an NB GREAT-layer is an attention sublayer, that utilizes attention scores $\alpha'$ and $\alpha''$ to compute a temporary node-feature $x$ for each node $i$ according to

$$x_i = \Big( \sum_{l \in E^+(i)} \alpha'_{il} \, W'_1 e_l \, || \sum_{l' \in E^-(i)} \alpha''_{il'} \, W''_1 e_{l'} \Big). \tag{7}$$

Here, $||$ is the concatenation operation, $e_l$ denotes an edge embedding for edge $l$, $W'_1$ and $W''_1$ are trainable weight matrices and $E^+(i)$, $E^-(i)$ denote outgoing and incoming edges respectively. New edge embeddings are then computed according to

$$e'_l = W_2 \left( x_{\text{start}(l)} \, || \, x_{\text{end}(l)} \right), \tag{8}$$

where $\text{start}(l)$ and $\text{end}(l)$ denote the start and end nodes of the directed edge $l$ and $W_2$ is another trainable matrix. The attention layers are arranged into a single GREAT-layer together with feed-forward layers, residual connections and normalization according to the original transformer architecture (Vaswani et al., 2017). Note that the residual connections are important as they allow parallel edges to have differing embeddings.

## 4.1 EDGE-BASED GMS

Vanilla node-based construction is insufficient for our purpose, as it relies on predicting the next node in the route, while we also require an edge-selection between the current node and the next. Thus, we propose an end-to-end edge-based model that instead predicts the next edge, and thereby implicitly the next node. We call this Edge-based GMS (GMS-EB).

We visualize GMS-EB in Figure 1. The encoder, consisting of $L$ GREAT-layers, outputs edge embeddings. Using them, the decoder constructs valid tours autoregressively. Given the instance $s$ and incomplete route $\pi_{1:t-1}$ in construction step $t$, the decoder selects edge $\pi_t$ with probability $p_{\theta(\lambda)}(\pi_t \mid \pi_{1:t-1}, s)$. Thus the probability of the whole route $\pi$ is

$$p_{\theta(\lambda)}(\pi \mid s) = p_{\theta(\lambda)}(\pi_1 \mid s) \prod_{t=2}^{T} p_{\theta(\lambda)}(\pi_t \mid \pi_{1:t-1}, s). \tag{9}$$

Here, $\theta(\lambda)$ is the weights of the model given the current preference $\lambda$. As Navon et al. (2021) and Lin et al. (2022), we utilize a Multi-Layer Perceptron (MLP) hyper-network to preference-condition the decoder, while keeping the encoder preference-agnostic. Consequently we form the decoder weights according to $\theta_{\text{dec}}(\lambda) = \text{MLP}(\lambda)$ and set $\theta(\lambda) = [\theta_{\text{enc}}, \theta_{\text{dec}}(\lambda)]$. Regarding the decoder architecture, it is an edge-based variant of the Multi-Pointer (MP) decoder from Jin et al. (2023). We detail this novel component further in Appendix C.

Note that a multigraph with $N$ nodes and $M$ edges between each pair of nodes has $\mathcal{O}(MN^2)$ edges in total. In our decoding, we only calculate scores for outgoing edges from each node, but all nodes will eventually be visited and thus all edge embeddings must be stored in memory. Moreover, the decoder must be run $\mathcal{O}(N)$ times to construct a full tour and we roll out $\mathcal{O}(N)$ sample trajectories according to the POMO framework (Kwon et al., 2020). Hence, an edge-based decoder scales as $\mathcal{O}(MN^4)$ compared to a node-based one that scales as $\mathcal{O}(N^3)$. Consequently, we propose a second architecture that employs a node-based decoder and thereby scales better to higher node counts in terms of memory requirements and runtime.

## 4.2 DUAL HEAD GMS

For a node-based decoder, we need to ensure that the action of choosing a node uniquely defines which edge to choose. This is achieved by pruning parallel edges, leaving a unique one between each node pair. In general, choosing this edge is not trivial. As such, we propose to learn the choice using a dual decoder setup.

We call this approach Dual Head GMS (GMS-DH) and it is visualized in Figure 2. The key is that we insert another decoder before the final GREAT layer, which uses intermediate edge embeddings to select the parallel edge to keep. Let $E_{ij}$ be the set of edges between the node pair $(i, j)$. The selection-decoder produces the probability $q_{\tilde{\theta}(\lambda)}(l \mid i, j, s)$ of retaining $l \in E_{ij}$. As we treat each node pair independently and select one edge per pair, the probability of the joint selection $\mathcal{E}$ of active edges is

$$q_{\tilde{\theta}(\lambda)}(\mathcal{E} \mid s) = \prod_{l \in \mathcal{E}} q_{\tilde{\theta}(\lambda)}(l \mid \text{start}(l), \text{end}(l), s). \quad (10)$$

Again, the weights $\tilde{\theta}(\lambda)$ are generated by an MLP hypernetwork. We detail the architecture for the selection-decoder further in Appendix C.

The edge choice from the first decoder is fed back into the encoder, where the embeddings of corresponding edges are aggregated into node embeddings using equation 7. After this final GREAT layer, we find it beneficial to add a small number of standard transformer layers to obtain more expressive node embeddings for the second decoder, which constructs routes.

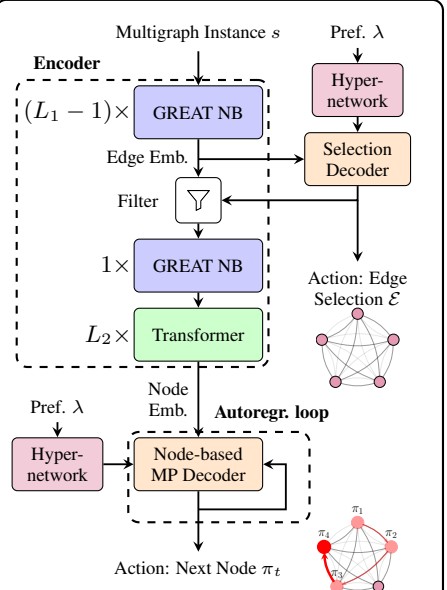

Figure 1: Edge-based GMS and its most important components.

The routing-decoder is a preference-conditioned version of the multi-pointer decoder (Jin et al., 2023). Similarly to the edge-based version above, given the partial tour $\pi_{1:t-1}$, this decoder outputs the probabilities $p_{\theta(\lambda)}(\pi_t \mid \pi_{1:t-1}, s, \mathcal{E})$ for the next construction step, where $\pi_t$ now denotes a node and the probability depends on the edge choice $\mathcal{E}$.

Remark that when applying GMS-DH to simple graphs, we simply remove the first decoder. Also note that this architecture ensures that the first $L_1 - 1$ layers only need to be run once for each instance as they are preference-agnostic and utilize the entire multigraph. This reduces the runtime compared to letting the encoder be preference-aware or pruning the multigraph for each preference before encoding.

From a broader perspective, our approach in GMS-DH can be viewed as a combination of Autoregressive (AR) and Non-Autoregressive (NAR) construction. As a drawback, the edge choice can no longer depend on the context given by the partial tour. Instead, it must be chosen before starting roll-out. Combinations of AR and NAR construction have previously been explored for divide-and-conquer approaches to tackle large-scale VRPs (Ye et al., 2024). However, this combination of NAR pruning and AR construction, and using a single joint encoder, is novel and specifically designed for the multigraph setting. We further clarify the technical novelty of the different components in GMS-DH, as well as GMS-EB, in Appendix B.3.

### 4.3 TRAINING ALGORITHM

Our first proposed model, GMS-EB, is trained using the Multi-Objective REINFORCE from Lin et al. (2022). Given a preference $\lambda \sim \Lambda$ and instance $s \sim S$, the model parameterizes a policy $p_{\theta(\lambda)}(\cdot \mid s)$ according to equation 9. A tour $\pi$ is associated with a Chebyshev reward corresponding to the negative scalarized cost equation 5:

$$R_\lambda(\pi) = -\max_i \{\lambda_i | f_i(\pi) - z_i^* |\}. \quad (11)$$

Figure 2: Dual head GMS and its most important components

The objective during training is to maximize

$$J(\theta) = \mathbb{E}_{\pi \sim p_\theta, \lambda \sim \Lambda, s \sim S}[R_\lambda(\pi)]. \quad (12)$$

To stabilize the training, we form a baseline $b_\lambda(s_i)$ by rolling out $K$ tours for the same instance $s_i$ and averaging the reward, according to the POMO framework (Kwon et al., 2020). The policy gradient is then utilized to update the model weights and it is approximated, given one preference $\lambda$, as

$$\nabla J(\theta) \approx \frac{1}{BK} \sum_{i,j=1}^{B,K} \left( R_\lambda(\pi_{ij}) - b_\lambda(s_i) \right) \nabla_\theta \log(p_{\theta(\lambda)}(\pi_{ij} \mid s_i)), \tag{13}$$

where $B$ is the batch size of instances.

For the second model, we propose a more involved training setup. Given an edge selection $\mathcal{E} \sim q_{\tilde{\theta}(\lambda)}$ from the selection-head according to equation 10, the routing-head parameterizes a policy $p_{\theta(\lambda)}(\cdot \mid s, \mathcal{E})$ and obtains a scalar reward $R_\lambda^{(2)}(\pi) = R_\lambda(\pi)$ by equation 11 as usual. With a fixed $\tilde{\theta}$, the objective for this head is to maximize

$$J_2(\theta) = \mathrm{E}_{\pi \sim p_\theta, \, \lambda \sim \Lambda, \, s \sim S, \, \mathcal{E} \sim q_{\tilde{\theta}}}[R_\lambda^{(2)}(\pi)]. \tag{14}$$

On the other hand, the task of the selection-head is to ensure that the edge choice is optimal. Let $\tilde{\pi}(\mathcal{E})$ denote the optimal tour on the pruned graph given selection $\mathcal{E}$. Then, define the reward and objective function based on the optimal tour according to

$$\begin{aligned}
R_\lambda^{(1)}(\mathcal{E}) &= R_\lambda(\tilde{\pi}(\mathcal{E})), \\
J_1(\tilde{\theta}) &= \mathrm{E}_{\lambda \sim \Lambda, \, s \sim S, \, \mathcal{E} \sim q_{\tilde{\theta}}}[R_\lambda^{(1)}(\mathcal{E})].
\end{aligned} \tag{15}$$

As the optimal tour $\tilde{\pi}$ is not available, we approximate it using $K_2$ sampled tours $\pi_1, \ldots, \pi_{K_2} \sim p_{\theta(\lambda)}(\cdot \mid s, \mathcal{E})$:

$$R_\lambda^{(1)}(\mathcal{E}) \approx \max_{k=1,\ldots,K_2} R_\lambda(\pi_k). \tag{16}$$

A baseline $b_\lambda^{(1)}(s_i)$ for the selection-head is formed by averaging $R^{(1)}$ over $K_1$ samples while a baseline $b_\lambda^{(2)}(s_i, \mathcal{E}_{ij})$ for the routing-head is formed by averaging over the $K_2$ samples for a fixed $s_i$ and $\mathcal{E}_{ij}$. Thus, we approximate the policy gradients with

$$\begin{aligned}
\nabla J_1(\tilde{\theta}) &\approx \frac{1}{BK_1} \sum_{i,j=1}^{B,K_1} \left( R_\lambda^{(1)}(\mathcal{E}_{ij}) - b_\lambda^{(1)}(s_i) \right) \nabla_{\tilde{\theta}} \log(q_{\tilde{\theta}}(\mathcal{E}_{ij} \mid s_i)), \\
\nabla J_2(\theta) &\approx \frac{1}{BK_1K_2} \sum_{i,j,k=1}^{B,K_1,K_2} \left( R_\lambda^{(2)}(\pi_{ijk}) - b_\lambda^{(2)}(s_i, \mathcal{E}_{ij}) \right) \nabla_\theta \log(p_\theta(\pi_{ijk} \mid s_i, \mathcal{E}_{ij})).
\end{aligned} \tag{17}$$

We outline the training of GMS-DH in Algorithm 1.

In addition, we utilize a form of curriculum learning to speed up training for both models. We start by training on small graph sizes and then gradually increase the problem size. For GMS-DH in the multigraph setting, we also start by training the routing-head on simple graphs to ensure that the approximation equation 16 is accurate even during the initial epochs of multigraph training. Further details regarding the curriculum learning can be found in Appendix D.

## 5 EXPERIMENTAL STUDIES

In this section we show the empirical performance for our two proposed methods on several routing problems across multiple instance sizes and distributions. We compare against evolutionary algorithms, neural baselines and state-of-the-art single-objective heuristics.

**Problems.** In the asymmetric simple graph case we consider two standard routing problems: The Multi-Objective Traveling Salesman Problem (MOTSP) and Multi-Objective Capacitated Vehicle Routing Problem (MOCVRP). Specifically, we consider bi-objective variants of these. In the MOTSP, we sample two distances for each node pair. These are summarized over the tour to yield the objective values. Regarding the distances, we consider both the TMAT (Kwon et al., 2021) and XASY distributions (Gaile et al., 2022), where the former obeys the triangle inequality and the latter does not - making it more difficult. For the MOCVRP, we sample one distance according to these

distributions. We then let the first objective be given by the total distance of a tour and the second objective by the makespan, which is the longest trip for a single vehicle. The Euclidean version of this problem has been used extensively for benchmarking in previous works (Lin et al., 2022).

In the multigraph setting, we consider the Multigraph MOTSP (MG-MOTSP) and Multigraph MOCVRP (MGMOCVRP) with two objectives, where we sample two-dimensional edge distances using distributions called FIX$x$ and FLEX$x$. Here, $x$ denotes the number of edges between each node pair. In the FLEX-distribution, we sample edge distances independently and remove edges which are dominated. In the FIX distribution we first sample edges independently, but then rearrange by sorting, so that the an edge

---

**Algorithm 1** One batch of REINFORCE for GMS-DH

**Input**: Preference distribution $\Lambda$, instance distribution $S$, batch size $B$, number of samples per instance head 1, $K_1$, and head 2, $K_2$

1: Sample preference $\lambda \sim \Lambda$
2: Sample instances $s_i \sim S$, $i = 1, \ldots, B$
3: Select edges $\mathcal{E}_{ij} \sim q_{\tilde{\theta}(\lambda)}(\cdot \mid s_i)$, $j = 1, \ldots, K_1$
4: Sample tours $\pi_{ijk} \sim p_{\theta(\lambda)}(\cdot \mid s_i, \mathcal{E}_{ij})$, $k = 1, \ldots, K_2$
5: Calculate $R_\lambda^{(1)}(\mathcal{E}_{ij})$, $R_\lambda^{(2)}(\pi_{ijk})$, $b_\lambda^{(1)}(s_i)$, $b_\lambda^{(2)}(s_i, \mathcal{E}_{ij})$
6: Calculate gradients $\nabla J_1(\tilde{\theta})$, $\nabla J_2(\theta)$ with equation 17
7: $\tilde{\theta} \leftarrow \textbf{ADAM}(\tilde{\theta}, \nabla J_1(\tilde{\theta}))$, $\theta \leftarrow \textbf{ADAM}(\theta, \nabla J_2(\theta))$,

---

that is best in one objective is the worst in the other objective. Thus, in the FIX$x$-distribution there are always exactly $x$ edges between each node pair, whereas FLEX$x$ has at most $x$ edges.

In Appendix E, we provide more in-depth explanations of the problems and distributions. We also provide results on Euclidean problems and tri-objective problems in Appendix H.

**Baselines.** For the MOTSP, MGMOTSP and MGMOCVRP, we utilize linear scalarization together with strong single-objective methods in the form of LKH3 (Helsgaun, 2017), Google OR-tools (Perron & Furnon, 2019) and HGS (Vidal et al., 2012). We also benchmark against four Multi-Objective Evolutionary Algorithms (MOEAs): MOGLS (Ishibuchi & Murata, 1998), MOEA/D (Zhang & Li, 2007), NSGA-II (Deb et al., 2002) and NSGA-III (Blank et al., 2019). Below, we only show the results for the MOEA that performs best in each scenario.

Finally, we also design a neural baseline based on the MatNet-architecture (Kwon et al., 2021) as encoder and a preference-conditioned MP network as decoder. We call this MatNet-Based Model (MBM) and our motivation is that it is a method that can be used almost "off-the-shelf" by combining components from literature. In the multigraph setting, this architecture requires us to pre-process the input by pruning edges until we obtain a simple graph, which we do using linear scalarization following Proposition 1.

In Appendix C and F, we provide further details about the neural benchmark and MOEAs respectively. In Appendix H, we also show results for all MOEAs as well as some other weaker heuristics.

**Model Settings**. We train all models for 200 epochs using 100 000 randomly generated instances per epoch with the ADAM optimizer (Kingma & Ba, 2015). We use the batch size $B = 64$ and the learning-rate $\eta = 10^{-4}$. For each batch, we sample a preference $\lambda = (\lambda_1, \lambda_2)$ using $\lambda_1 \sim \text{Unif}[0, 1]$ and $\lambda_2 = 1 - \lambda_1$. For GMS-EB we let the number of layers be $L = 6$, while GMS-DH has $L_1 = 5$ and $L_2 = 2$. Both models have embedding dimension 128 and use 8 attention heads in the GREAT layers and MP decoders. Regarding the number of samples in the decoding, we set $K_1 = 4$ for the selection-decoder during training and $K_1 = 1$ during inference. In the routing-decoder, we always set $K_2$ to the problem size as customary in the POMO framework (Kwon et al., 2020). Finally, we augment the models during inference with a factor of 8. For MBM, this is done using the original MatNet-augmentation (Kwon et al., 2021), while for GMS we use the scaling-augmentation of Lischka et al. (2025).

**Metrics**. We utilize the Hypervolume (HV) metric (Zitzler et al., 2003) to evaluate the performance of the methods. Formally, for a given set of points $S = \{y_1, \ldots, y_n\} \subset \mathbb{R}^m$ and a reference point $r \in \mathbb{R}^m$ that is dominated by all points in $S$, the HV is defined as:

$$\text{HV}(S, r) = \Lambda \left( \bigcup_{i=1}^{n} [y_i, r] \right), \tag{18}$$

where $\Lambda$ denotes the Lebesgue measure in $\mathbb{R}^m$, and $[y_i, \mathbf{r}]$ represents the axis-aligned hyperrectangle bounded by $y_i$ and $r$. Intuitively, a higher HV generally indicates better performance, since the points in the obtained Pareto front are further from the reference.

We report the normalized HV, which is scaled to lie in the interval $[0, 1]$, averaged over 200 test instances. For the neural models and benchmarks relying on scalarization, we use 101 preferences linearly spaced between $\lambda = (1, 0)$ and $(0, 1)$ during inference to obtain the Pareto front. Apart from the HV, we provide the HV gap compared to the best-performing method as well as the inference time. In Appendix D, we also report some training times for the learning-based methods.

**Hardware.** We conducted training and inference for the learning-based methods using a single NVIDIA Tesla A40 GPU with 48GB of VRAM. For the non-learning-based methods, evaluations were performed on an Intel Xeon Gold 6338 CPU and, to enable a more fair comparison, we report execution time based on solving multiple instances in parallel. Akin to Kwon et al. (2021), we set the number of parallel processes to 8.

## 5.1 PERFORMANCE ON BENCHMARK PROBLEMS

We show the performance of GMS together with the other methods in Table 1. Both variants of GMS outperform most benchmarks on all problems. On the MOCVRP they are the best performing methods, while only LKH outperform them on the MOTSP and MGMOTSP. For the MGMOCVRP, HGS attains the highest hypervolume, with GMS-EB and GMS-DH following in all cases except FIX2-50, where MBM performs better than GMS-DH. This showcases that our architectures are effective across a variety of distributions, instance sizes and problems. The baseline neural method, MBM, also performs quite well across many cases, especially on the MGMOTSP and MGMOCVRP, but severely underperforms on some distributions, e.g., XASY100 for the MOCVRP. In Appendix H, we include results for other naive extensions of existing neural methods, and show that they also tend to be less effective and robust compared to GMS-EB and GMS-DH.

However, MBM is usually slightly faster than GMS-DH, owing to the MatNet architecture being faster and more memory-efficient than GREAT (for a direct comparison, see the results by Lischka et al. (2025)). GMS-DH has a small advantage for the MGMOTSP with 50 nodes due to only encoding each multigraph once, but this is offset again for 100 nodes. Nonetheless, both GMS models remain efficient relative to the other benchmarks, despite the slower scaling of GMS-EB. In particular, GMS-DH as well as GMS-EB without augmentation are significantly faster than LKH and HGS.

For the purpose of ablation, apart from the previously mentioned benchmarks, we also evaluate a variant of GMS-DH with Pre-Pruning (GMS-DH PP) applied before the encoder in the MGMOTSP and MGMOCVRP settings. This pre-pruning is the same as for MBM, and the selection head is removed accordingly.

The poor performance of GMS-DH PP in Table 1 highlights the importance of explicitly designing for the multigraph structure, even when a theoretically sound pruning strategy is available. Since GMS-DH PP requires re-running the encoder for each preference, it is approximately $3\times$ slower than GMS-DH on the MGMOTSP and $2\times$ slower on the MGMOCVRP. We also hypothesize that the pre-pruning step shifts the data distribution in a way that GMS-DH is ill-equipped to model, resulting in performance that is worse than MBM with the same pruning strategy. Indeed, in our experience GMS-DH PP tends to produce Pareto fronts with poor diversity, clustered around central preference regions.

## 5.2 ZERO-SHOT GENERALIZATION

We also test our models on larger instances not seen during training, and report the results in Table 2. Both GMS-DH and GMS-EB remain competitive across both problems and all distributions in this zero-shot setting. Note that we refrain from testing the MOEAs here, as they already underperform significantly for smaller instances in Table 1. Moreover, MatNet is not applicable since it is limited by its embedding dimension (set to 128 in our case).

Table 1: MOTSP, MOCVRP, MGMOTSP and MGMOCVRP with different instance sizes (50, 100) and distributions. Runtime is the total time for solving 200 instances. The best results (excluding LKH and HGS, our baselines) are in **bold** and the second best are underlined. Remark that most neural MO solvers benchmark on instances of roughly this size (Chen et al., 2023b; Fan et al., 2025a).

**MOTSP**

| | TMAT50 HV | Gap | Time | TMAT100 HV | Gap | Time | XASY50 HV | Gap | Time | XASY100 HV | Gap | Time |
|---|---|---|---|---|---|---|---|---|---|---|---|---|
| LKH | 0.58 | 0.00% | (6.4m) | 0.63 | 0.00% | (29m) | 0.83 | 0.00% | (6.9m) | 0.90 | 0.00% | (32m) |
| OR Tools | 0.54 | 6.13% | (29m) | 0.59 | 7.12% | (2.5h) | 0.77 | 7.25% | (24m) | 0.85 | 6.06% | (1.8h) |
| MOEA/D | 0.53 | 9.51% | (3.7h) | 0.50 | 20.9% | (13h) | 0.73 | 12.68% | (3.6h) | 0.70 | 22.22% | (13h) |
| MBM | 0.50 | 13.5% | (3.9s) | 0.56 | 11.2% | (19s) | 0.75 | 10.37% | (3.7s) | 0.84 | 7.41% | (19s) |
| MBM (aug) | 0.52 | 10.2% | (27s) | 0.58 | 9.17% | (2.4m) | 0.76 | 8.28% | (27s) | 0.85 | 6.34% | (2.4m) |
| GMS-EB | 0.57 | 1.50% | (25s) | 0.63 | 1.45% | (3.6m) | 0.81 | 2.94% | (25s) | 0.88 | 3.03% | (3.6m) |
| GMS-EB (aug) | **0.57** | **1.14%** | (3.3m) | **0.63** | **1.13%** | (29m) | **0.82** | **2.06%** | (3.3m) | **0.88** | **2.76%** | (28m) |
| GMS-DH | 0.57 | 2.36% | (4.1s) | 0.62 | 2.76% | (20s) | 0.80 | 4.12% | (3.9s) | 0.86 | 4.87% | (20s) |
| GMS-DH (aug) | 0.57 | 1.76% | (28s) | 0.62 | 2.19% | (2.6m) | 0.81 | 2.84% | (28s) | 0.87 | 3.85% | (2.6m) |

**MOCVRP**

| | TMAT50 HV | Gap | Time | TMAT100 HV | Gap | Time | XASY50 HV | Gap | Time | XASY100 HV | Gap | Time |
|---|---|---|---|---|---|---|---|---|---|---|---|---|
| MOEA/D | 0.41 | 13.98% | (24h) | 0.19 | 58.70% | (76h) | 0.65 | 13.88% | (24h) | 0.42 | 47.23% | (83h) |
| MBM | 0.47 | 1.33% | (6.2s) | 0.43 | 4.50% | (27s) | 0.70 | 6.57% | (6.2s) | 0.60 | 24.47% | (27s) |
| MBM (aug) | 0.47 | 0.99% | (41s) | 0.44 | 3.29% | (3.3m) | 0.72 | 4.13% | (42s) | 0.64 | 19.60% | (3.4m) |
| GMS-EB | 0.47 | 0.25% | (36s) | 0.45 | 0.57% | (4.2m) | 0.74 | 1.89% | (36s) | 0.79 | 1.23% | (4.3m) |
| GMS-EB (aug) | **0.47** | **0.00%** | (5.0m) | **0.45** | **0.00%** | (33m) | **0.75** | **0.00%** | (4.9m) | **0.80** | **0.00%** | (48m) |
| GMS-DH | 0.47 | 0.76% | (6.8s) | 0.45 | 1.46% | (32s) | 0.72 | 4.13% | (6.6s) | 0.77 | 3.60% | (33s) |
| GMS-DH (aug) | 0.47 | 0.36% | (47s) | 0.45 | 0.79% | (4.3m) | 0.74 | 1.65% | (47s) | 0.79 | 1.21% | (4.7m) |

**MGMOTSP**

| | FLEX2-50 HV | Gap | Time | FLEX2-100 HV | Gap | Time | FIX2-50 HV | Gap | Time | FIX2-100 HV | Gap | Time |
|---|---|---|---|---|---|---|---|---|---|---|---|---|
| LKH | 0.90 | 0.00% | (6.3m) | 0.94 | 0.00% | (31m) | 0.92 | 0.00% | (6.3m) | 0.95 | 0.00% | (30m) |
| OR Tools | 0.86 | 4.21% | (24m) | 0.91 | 3.46% | (1.8h) | 0.90 | 2.59% | (24m) | 0.93 | 2.25% | (2.0h) |
| MOEA/D | 0.74 | 17.50% | (18h) | 0.68 | 28.46% | (99h) | 0.78 | 15.42% | (18h) | 0.72 | 24.69% | (91h) |
| MBM | 0.86 | 5.21% | (13s) | 0.91 | 3.72% | (44s) | 0.90 | 2.46% | (13s) | 0.93 | 2.41% | (44s) |
| MBM (aug) | 0.87 | 3.98% | (1.7m) | 0.91 | 3.28% | (5.9m) | 0.91 | 1.66% | (1.7m) | 0.93 | 2.10% | (5.9m) |
| GMS-EB | 0.88 | 2.00% | (56s) | 0.93 | 1.81% | (9.1m) | 0.91 | 1.27% | (57s) | 0.94 | 1.33% | (9.2m) |
| GMS-EB (aug) | 0.89 | 1.50% | (7.5m) | **0.93** | **1.47%** | (1.2h) | **0.91** | **1.02%** | (7.5m) | **0.94** | **1.09%** | (1.2h) |
| GMS-DH | 0.89 | 1.84% | (13s) | 0.92 | 2.11% | (49s) | 0.91 | 1.73% | (12s) | 0.93 | 2.13% | (52s) |
| GMS-DH (aug) | **0.89** | **1.45%** | (1.5m) | 0.93 | 1.61% | (6.6m) | 0.91 | 1.40% | (1.5m) | 0.93 | 1.92% | (6.6m) |
| GMS-DH PP | 0.77 | 14.24% | (33s) | 0.83 | 12.53% | (2.3m) | 0.82 | 10.51% | (33s) | 0.85 | 10.39% | (2.3m) |

**MGMOCVRP**

| | FLEX2-50 HV | Gap | Time | FLEX2-100 HV | Gap | Time | FIX2-50 HV | Gap | Time | FIX2-100 HV | Gap | Time |
|---|---|---|---|---|---|---|---|---|---|---|---|---|
| HGS | 0.89 | 0.00% | (15h) | 0.89 | 0.00% | (75h) | 0.87 | 0.00% | (15h) | 0.91 | 0.00% | (74h) |
| OR Tools | 0.84 | 5.93% | (1.5h) | 0.84 | 5.97% | (1.9h) | 0.83 | 5.14% | (1.5h) | 0.88 | 3.78% | (1.9h) |
| NSGA-II | 0.71 | 20.05% | (37h) | 0.56 | 36.54% | (219h) | 0.65 | 25.16% | (36h) | 0.61 | 33.28% | (224h) |
| MBM | 0.83 | 6.07% | (16s) | 0.81 | 9.38% | (54s) | 0.84 | 4.14% | (16s) | 0.87 | 5.00% | (54s) |
| MBM (aug) | 0.84 | 5.23% | (2.0m) | 0.82 | 8.30% | (6.9m) | 0.84 | 3.39% | (1.9m) | 0.87 | 4.32% | (6.9m) |
| GMS-EB | 0.85 | 4.12% | (1.2m) | 0.85 | 4.08% | (10m) | 0.84 | 3.47% | (1.2m) | 0.89 | 2.81% | (9.9m) |
| GMS-EB (aug) | **0.86** | **3.55%** | (9.0m) | **0.86** | **3.66%** | (1.4h) | **0.85** | **2.71%** | (11m) | **0.89** | **2.38%** | (1.4h) |
| GMS-DH | 0.85 | 4.86% | (20s) | 0.85 | 4.94% | (1.0m) | 0.83 | 4.96% | (20s) | 0.88 | 3.58% | (1.0m) |
| GMS-DH (aug) | 0.85 | 3.86% | (2.1m) | 0.85 | 3.89% | (8.2m) | 0.84 | 3.96% | (2.1m) | 0.88 | 3.05% | (8.1m) |
| GMS-DH PP | 0.75 | 15.60% | (36s) | 0.75 | 15.70% | (2.45m) | 0.76 | 12.82% | (36s) | 0.81 | 11.05% | (2.45m) |

## 5.3 A DIFFICULT MULTIGRAPH PROBLEM

Finally, we revisit one of the main motivations behind this work: routing for problems without clear a priori edge selection. Thus, we evaluate the performance on a more complex problem. Specifically, inspired by Ben Ticha et al. (2017), we consider an MGMOTSP with Time-Windows (MGMOTSPTW). Each edge is associated with a time and distance, while the objectives correspond to the number of violated time-windows and the total distance. Remark that, unlike the MGMOTSP and MGMOCVRP, it is not clear in this case which edges can be pruned due to the more intricate time-window objective.

Besides MBM with linear pre-pruning and MOEAs, we benchmark against a simplified version of GMS-DH in order to ablate its ability to learn which edges are optimal. In this simplified version, we replace the selection-decoder with a simple function that removes edges with suboptimal linear cost. Compared to pre-pruning the multigraph before the encoder, this approach is smarter, as it removes the need to reencode. Furthermore, this simple variant actually performs quite well on the MGMOTSP, which we show and discuss in Appendix G.

We display the results for MGMOTSPTW in Table 3. Our two models are the best performing ones by quite a large margin. Notably, vanilla GMS-DH also outperforms the simplified variant, which illustrates the value of learning to select beneficial edges in comparison to using a simple pruning heuristic.

Table 2: MOTSP and MGMOTSP for larger instances (zero-shot generalization). Performance over 100 instances.

| | | TMAT200 | | | XASY200 | | |
|---|---|---|---|---|---|---|---|
| | | HV | Gap | Time | HV | Gap | Time |
| **MOTSP** | LKH | 0.67 | 0.00% | (1.2h) | 0.95 | 0.00% | (1.3h) |
| | OR Tools | 0.62 | 7.89% | (6.5h) | 0.90 | 4.46% | (4.0h) |
| | GMS-EB | **0.66** | **1.86%** | (17m) | **0.92** | **2.78%** | (17m) |
| | GMS-DH | 0.65 | 3.59% | (1.0m) | 0.90 | 5.22% | (1.0m) |
| | | FLEX2-200 | | | FIX2-200 | | |
| **MGMOTSP** | LKH | 0.97 | 0.00% | (1.2h) | 0.97 | 0.00% | (1.2h) |
| | OR Tools | 0.94 | 2.53% | (3.9h) | 0.95 | 1.73% | (4.0h) |
| | GMS-EB | **0.95** | **1.74%** | (50m) | **0.96** | **1.34%** | (50m) |
| | GMS-DH | 0.95 | 2.27% | (2.2m) | 0.94 | 2.56% | (2.2m) |

Table 3: MGMOTSPTW on the FLEX2 and FIX2 distributions with 50 nodes. Performance over 200 instances. In GMS-DH Simple, the selection-decoder is replaced with a simple pruning function.

| | FLEX2-50 | | | FIX2-50 | | |
|---|---|---|---|---|---|---|
| | HV | Gap | Time | HV | Gap | Time |
| MOEA/D | 0.60 | 32.79% | (34h) | 0.60 | 35.73% | (36h) |
| MBM | 0.80 | 11.09% | (14s) | 0.64 | 31.18% | (14s) |
| MBM (aug) | 0.81 | 10.10% | (1.8m) | 0.66 | 28.58% | (1.9m) |
| GMS-EB | 0.89 | 0.97% | (60s) | 0.93 | 0.55% | (59s) |
| GMS-EB (aug) | **0.90** | **0.00%** | (7.9m) | **0.93** | **0.00%** | (7.9m) |
| GMS-DH | 0.85 | 5.10% | (13s) | 0.91 | 1.74% | (12s) |
| GMS-DH (aug) | 0.87 | 2.69% | (1.6m) | 0.92 | 0.78% | (1.7m) |
| GMS-DH Simple | 0.83 | 7.01% | (13s) | 0.85 | 8.93% | (12s) |

## 6 CONCLUSION

In this paper, we propose two learning-based approaches for solving multi-objective routing problems, such as the MOTSP and MOCVRP, on multigraphs and asymmetric graphs. Both methods utilize graph neural networks for encoding the graph structure but differ in their route construction strategies. The first method is edge-based and directly builds tours on the multigraph, while the second employs a two-stage decoding process, consisting of non-autoregressive pruning of the multigraph followed by route construction.

Empirical results show that both methods perform well across a variety of settings. While the edge-based model is simpler and usually demonstrates slightly superior performance, the second model is more scalable due to lower memory and runtime demands. In the future, we will explore alternative decomposition methods to pruning, to potentially obtain even more scalable methods.

Finally, we highlight that our models can be valuable in all routing scenarios where multigraphs are appropriate, i.e., when there are multiple competing edges between each node pair. In future work, we plan to use our setup to address single-objective problems with hard constraints as well as stochastic variants.

ACKNOWLEDGEMENTS

This work was performed as a part of the research project "LEAR: Robust LEArning methods for electric vehicle Route selection" funded by the Swedish Electromobility Centre (SEC). The computations were enabled by resources provided by the National Academic Infrastructure for Supercomputing in Sweden (NAISS) at Chalmers e-Commons partially funded by the Swedish Research Council through grant agreement no. 2022-06725. Through the affiliation of F.R., the work was also partially supported by the Wallenberg AI, Autonomous Systems and Software Program (WASP) funded by the Knut and Alice Wallenberg Foundation.

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

# A  EXTENDED RELATED WORK

Here, we complement the literature review in the main text with a more thorough discussion of related work.

## A.1  SINGLE-OBJECTIVE LEARNING FOR ROUTING

In the autoregressive paradigm, Vinyals et al. (2015) were early to propose learning solutions to the TSP. They utilized an oracle solver for supervised learning, and Bello et al. (2017) later introduced reinforcement learning for training. By using attention models and inherent symmetries in the problem, first Kool et al. (2018) and then Kwon et al. (2020) obtained greatly improved optimality gaps on various routing problems. Regarding asymmetric problems, Kwon et al. (2021) proposed Mat-Net, a mixed-attention architecture working on bi-partite graphs. More recent works are Jin et al. (2023); Drakulic et al. (2023); Chen et al. (2024); Bi et al. (2024). These papers have improved results, can handle constraints better and can scale to even larger problems than previously. Another recent notable contribution was made by Drakulic et al. (2025), who proposed a jointly trained generalist model to solve a multitude of combinatorial optimization problems, including examples from routing.

In the non-autoregressive category, some works are Joshi et al. (2019); Fu et al. (2021); Qiu et al. (2022); Sun & Yang (2023). Typically, these methods require more sophisticated procedures than simple sampling in the decoding, such as Monte–Carlo tree search, to perform competitively, but scale better to large instances due to lower space and time complexity. Thus, an interesting recent work combining autoregressive and non-autoregressive construction to solve large instances (up to $10^5$ nodes for the TSP) while maintaining performance is Ye et al. (2024).

Besides constructive approaches, *improvement-based* methods usually obtain great results, but with longer running-times, by refining initial solutions. Some examples are Chen & Tian (2019); Lu et al. (2020); Hottung & Tierney (2020); Hudson et al. (2022).

## A.2  MULTI-OBJECTIVE LEARNING FOR ROUTING

Early works treating MO routing using learning utilize multiple models, where one model is responsible for one preference between the objectives (Li et al., 2021; Wu et al., 2020). This quickly becomes infeasible as the number of models required grows exponentially with the number of objectives (Ehrgott, 2005). Pareto-set learning for multi-objective routing using a single model was proposed by Lin et al. (2022). They used the hyper-networks of Navon et al. (2021) to condition the decoder on the preference, which ensures the encoder only needs to be run once per instance. Recent studies that feature improved results or extend to more intricate VRP variants with a similar conditioning method include Li et al. (2023), Lu et al. (2024), Wu et al. (2024), Wu et al. (2025) and Fan et al. (2025a). An alternative approach is to use meta-learning to train a single model that can be quickly fine-tuned over a few steps to solve individual subproblems. Two representative works in this direction are Zhang et al. (2021) and Chen et al. (2023a).

Notably, all these prior methods utilize transformer-based architectures and operate on node coordinates, ensuring they are not suitable for asymmetric problems or multigraph problems. Besides tackling asymmetric problems, our work is novel in the MO routing setting as most methods utilize autoregressive construction, while one of our models combine autoregressive and non-autoregressive construction.

Another significant recent work is that of Chen et al. (2023b), in which two diversity enhancement methods, together labeled NHDE, are introduced. The authors combine NHDE with both the meta-learning approach and hyper-network approach to achieve improved results. Similarly, Fan et al. (2025b) develop a model-agnostic framework, POCCO, for improved training using pairwise preference learning and conditional computation blocks. Integrating GMS with NHDE or POCCO is possible in principle, but outside the scope of this paper.

### A.3 ROUTING ON MULTIGRAPHS

We note that apart from vehicle routing problems, which are the focus in this article, much research has been dedicated to MO shortest path problems on multigraphs. Successful approaches for these problems include genetic algorithms (Beke et al., 2021), A*-related methods (Weiszer et al., 2020) and hybrid approaches combining A* with learning (Liu et al., 2022). These methods all aim to find a Pareto set of shortest paths between two destinations.

## B EXTENDED MOTIVATION

In this section, we further motivate our work by explaining why most existing methods are unsuitable for routing on multigraphs. We also explain why transforming the underlying multigraph and applying vanilla node-based construction has significant scaling issues and other disadvantages. Finally, we discuss the technical novelty of the various components in GMS-DH and GMS-EB.

### B.1 NEURAL ROUTING METHODS EXTENDED TO MULTIGRAPHS

There exists many single-objective neural methods for routing capable of encoding simple asymmetric graphs (Kwon et al., 2021; Sun & Yang, 2023; Lischka et al., 2025; Drakulic et al., 2025). However, a significant amount of these, in particular transformer-based architectures like MatNet (Kwon et al., 2021) and GOAL (Drakulic et al., 2025), are unsuitable for encoding multigraphs. Both use distance-matrices as input to mixed attention blocks. Representing multigraphs through distance matrices ($N \times N \times D$ dimensions) by e.g., concatenation along the feature dimension will result in violated permutation invariance among parallel edges, which is key to maintain. As such, to extend these transformer-based encoders to multigraph inputs, one would need to design edge-permutation invariant attention layers, which to our knowledge has not been attempted. Note that we apply MatNet to multigraphs by pre-pruning so that there is only one edge between each node pair. This works, but learned pruning after encoding is more suitable in the general case when good pruning heuristics might not be available.

Additionally, some recent methods (Lischka et al., 2025; Meng et al., 2025) are edge-based and can encode multigraphs, but rely on node-selection in the decoding, ensuring they are not applicable to routing on multigraphs. An even smaller subset of related work, e.g., Zhou et al. (2025) in the autoregressive paradigm and Sun & Yang (2023) in the non-autoregressive paradigm, provide edge selections in the decoding, which would in theory make them applicable to multigraph routing. However this extension has not been attempted in any previous work we are aware of, and we hypothesize that direct applications would lead to problems akin to GMS-EB (which can be viewed as an initial extension of GREAT (Lischka et al., 2025)), where the scaling is slow due to not explicitly designing for the multigraph structure.

### B.2 NODE-BASED CONSTRUCTION AND GRAPH TRANSFORMATIONS

Firstly, it is possible to translate a multigraph into a simple graph by using its line graph, which is shown on the left side of Figure 3. This entails replacing each edge with a node and connecting those new nodes which share endpoints. In our context of fully connected graphs with $M$ parallel edges, this results in $\mathcal{O}(MN^2)$ nodes in the line graph and $\mathcal{O}(M^2N^3)$ edges. By utilizing node-based construction (that is, selecting one node at a time sequentially) on the new graph with $N$ POMO samples and in $N$ steps, the complexity in the decoding would thus be $\mathcal{O}(MN^4)$.

Another, related, approach is to introduce virtual end-point nodes for each parallel edge incident to a specific node, which is shown on the right side of Figure 3. Again, this leads to a higher node count, as each edge in the original graph is represented using one virtual node, yielding $\mathcal{O}(MN^2)$ total nodes. While the complexity of the total number of edges is lower in this case ($\mathcal{O}(MN^2)$, since we introduce only one edge per virtual node), for standard node-based decoding the complexity remains $\mathcal{O}(MN^4)$.

One can note that both these transformations combined with node-based construction resemble GMS-EB in terms of complexity and underlying mechanism. At its core, GMS-EB treats each edge as a node in the decoding. However, as GMS-EB works on the original graph, we believe it to

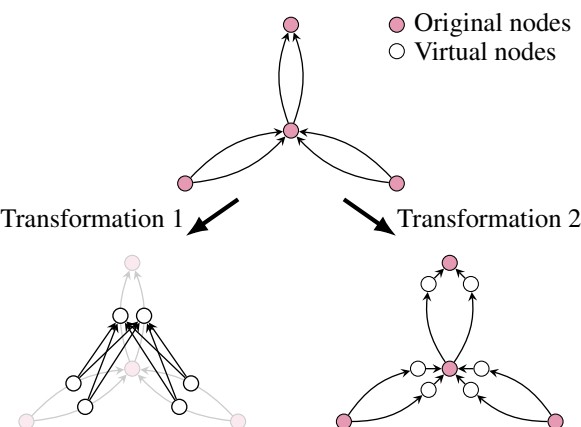

Figure 3: Two possible graph transformations to convert multigraphs into simple graphs. For simplicity, we show both transformations on a graph that is not fully connected.

be significantly more interpretable. Moreover, GMS-DH obtains lower complexity ($\mathcal{O}(N^3)$) in its decoding, since it is node-based in the original graph.

In summary, while it is possible to work on transformed simple graphs instead of underlying multigraphs, we obtain significant advantages by not doing so. Optimizing architectures on transformed graphs constitutes another research direction that would be interesting to pursue, but is beyond the scope of this work. As shown by the occasionally poor performance of MBM on asymmetric simple graphs in Table 1, it seems unlikely that such an "off-the-shelf" architecture would work well on much larger graphs induced by transforming multigraphs.

### B.3 TECHNICAL NOVELTY

Besides being the first and only neural approach to tackle multigraph routing problems, GMS is novel in several other aspects. Firstly, GMS-EB is quite novel in its edge-selection procedure during decoding. Some earlier works in the autoregressive paradigm, e.g., Zhou et al. (2025), utilize edge selection in the decoding, but not in the context of multigraphs to differentiate between edges between the same node pair. The exact edge-based multi-pointer decoder for GMS-EB, described in Appendix C.1, is also novel, being an extension of the light-weight multi-pointer decoder from Jin et al. (2023). Moreover, our edge-based decoder utilizes weighted scalarized costs in the scoring to obtain good initial tour generation, which to our knowledge is a novel addition in the multi-objective context.

Regarding GMS-DH, the combined Non-Autoregressive (NAR) and Autoregressive (AR) nature is novel in the edge-pruning and node permutation context, and specifically designed to avoid much of the scaling issues inherent with multigraphs. Previously, e.g., by Ye et al. (2024), NAR and AR model combinations have been explored for decomposing large graphs using divide-and-conquer strategies. Also note that we utilize the same encoder for both the NAR-part and AR-part of the model in the dual-decoder setup, which is atypical (Ye et al., 2024; Pan et al., 2025) and indicates that our encoder produces expressive edge embeddings. Additionally, the collaborative RL-based training of the two heads (outlined in Section 4.3) is different compared to other dual-decoder work (Bi et al., 2024) and NAR + AR work (Ye et al., 2024). It ensures all parts of the model can be trained simultaneously without access to high-quality labels. Finally, the selection-decoder, described in Appendix C.2, has been designed for our particular pruning task of choosing between parallel edges given a preference. Consequently, it has not been studies before in previous work.

## C ARCHITECTURE DETAILS

Here, we highlight some of the main components in the model architectures that are not described in the main text.

## C.1 EDGE-BASED DECODER IN GMS-EB

In GMS-EB, we utilize an edge-based multi-pointer decoder which is inspired by Jin et al. (2023). Its mathematical formulation is the following. Given trainable weight matrices $W_1, W_2, W_3, W_4$ we form the query, representing the current routing context, as

$$q_t = W_1 e_{\pi_1} + W_2 e_{\pi_{t-1}} + \frac{1}{N}(W_3 e_{\text{graph}} + W_4 e_{\text{visited}}). \tag{19}$$

Here, $e$ denotes an edge-embedding from the encoder, $\pi_1$ is the first visited edge and $\pi_{t-1}$ the most recently visited edge. Moreover, we utilize an enhanced context in which $e_{\text{graph}}$ is the sum of all embeddings in the graph and $e_{\text{visited}}$ is the sum over the visited edges according to

$$e_{\text{visited}} = \sum_{t'=1}^{t-1} e_{\pi_{t'}}. \tag{20}$$

Let $H$ be the number of heads/pointers. The score for edge $l$ is formed according to

$$\alpha_l = \frac{1}{H} \sum_{h=1}^{H} \frac{1}{\sqrt{d_k}} (W_h^q q_t)^T (W_h^k k_l) - \text{Scalar cost}(l). \tag{21}$$

Note that the key $k_l$ is the embedding $e_l$ for the edge and $d_k$ is its dimension. Similarly to the original article, we subtract a cost to speed up the exploration and ensure that costly edges are considered less favorable from the start. In our case, the cost is multi-objective so we must scalarize it. We find that both Chebyshev and linear scalarization work well for this purpose. The last step is to form the logits as

$$u_l = \begin{cases} c \tanh \alpha_l & \text{if } l \text{ connects current and unvisited node} \\ -\infty \end{cases} \tag{22}$$

Finally, softmax is applied to transform the logits to probabilities.

In our case, we also preference-condition the decoder using a hyper-network. This is done by letting a multi-layer perceptron generate the weights $W_1, W_2, W_3, W_4$ and $(W_h^q, W_h^k)_{h=1}^H$.

## C.2 SELECTION-DECODER IN GMS-DH

The task of this decoder is to prune parallel edges in the intermediate stage of GMS-DH, leaving only one between each node pair. Initially, we apply $L_3$ GREAT NB layers on the edge embeddings from the encoder to obtain embeddings specialized for this task. In the experiments we set $L_3 = 2$. These layers are not preference conditioned and hence they do not need to be rerun for each preference. Next, given two nodes, we want to compute the probability of retaining each (directed) edge in between. For this purpose, we essentially utilize a multi-pointer score calculation similar to equation 21. Given a node pair $(i, j)$, and the set of edges in between, $E_{ij}$, let the query be the average embedding according to

$$q_{ij} = \frac{1}{|E_{ij}|} \sum_{l \in E_{ij}} e_l. \tag{23}$$

Moreover, let the key $k_l$ be the embedding $e_l$. Then, form the score for edge $l$ as

$$\alpha_l = \frac{1}{H} \sum_{h=1}^{H} \frac{1}{\sqrt{d_k}} (W_h^q q_{ij})^T (W_h^k k_l) - \text{Scalar cost}(l), \tag{24}$$

where $i = \text{start}(l)$ and $j = \text{end}(l)$. Again, note that we subtract a scalar cost to improve exploration. Also note that $(W_h^q, W_h^k)_{h=1}^H$ are generated by a hyper-network. Final edge probabilities are then computed using tanh-clipping and softmax as

$$u_l = c \tanh(\alpha_l),$$
$$p_l = \frac{\exp(u_l)}{\sum_{l' \in E_{ij}} \exp(u_{l'})}. \tag{25}$$

### C.3 MATNET-BASED BENCHMARK MODEL

We visualize this architecture in Figure 4. The encoder consists of $L$ MatNet-layers (Kwon et al., 2021), while the decoder consists of an MP decoder (Jin et al., 2023) preference-conditioned with a hyper-network.

The MatNet-architecture operates on bipartite graphs. It utilizes mixed-score attention layers together with fully-connected layers and normalization as in standard transformer layers (Vaswani et al., 2017). These attention layers "mix" internally computed attention scores with external information from e.g., a distance matrix $D$. To accommodate the use of many edge features represented by distance matrices, $D_1, \ldots, D_f$, in our multi-objective setting, we follow the idea in Appendix A of the original paper. That is, we mix the internal attention score with all distance matrices using an MLP with $f + 1$ inputs.

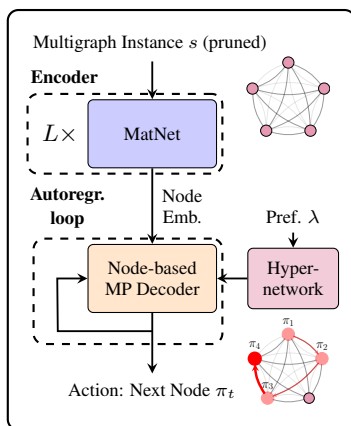

For the node-input to the initial MatNet-layer, in the absence of node-features we use zero-vectors for the "row" nodes and one-hot vectors for the "column" nodes, as in the original paper. If there are node features, e.g., demand in the MOCVRP, we use these as "row" input while keeping the one-hot "column" input. This allows us to augment each instance by sampling several one-hot combinations rather than scaling the instance as is done for GMS.

Lastly, we note that MatNet is not well-defined on multigraphs due to the fact that the distance matrices $D_1, \ldots, D_f$ represent unique edge features between nodes. Thus, we must pre-prune a multigraph before passing it through the encoder. This can be done with some sort of heuristic in the general case, but under linear scalarization and considering the MOTSP, an optimal pruning method is to remove edges with suboptimal

Figure 4: MatNet-Based Model (MBM).

scalarized cost. We utilize this pruning method for both multigraph problems. It can then be noted that the encoder needs to be rerun for each new preference, which results in a slower architecture than if the encoding is done once.

## D MODEL TRAINING

Here we outline the curriculum learning which we briefly discussed in the main text. We also present an analysis of training performance.

### D.1 CURRICULUM LEARNING DETAILS

For both GMS-EB and GMS-DH we utilize a simple curriculum learning approach, with the purpose to decrease training times and to improve performance on large problems. The idea is simple: we train on small instances before training on large instances with the same distribution.

For instances with 20 nodes, we do not use any such curriculum. For instances with 50 nodes we start by training for 100 epochs on 20 nodes and then train on 50 nodes for 100 epochs. Finally, for instances with size 100, we start by training for 100 epochs on size 20. Then, for GMS-DH, we train for 50 epochs on size 50 and 50 epochs on size 100. As GMS-EB takes quite long to train for 100 nodes, we are satisfied with training for 100 epochs on size 50 for this model and then fine-tune it for 10 epochs on size 100, resulting in 210 epochs in total. Remark that our curriculum is quite arbitrary, but the model performance is not contingent on this exact setup.

Additionally, we start by training GMS-DH for 5 epochs (included in the first 100 epochs) on the XASY distribution with 20 nodes. This is because the selection-decoder relies on a reward signal from the routing-decoder. As such, we find that performance and rate of convergence improves if the routing-decoder is pre-trained on simple graphs to obtain a stable and accurate signal. We also find that stability increases if we freeze the parameters of the shared encoder with respect to the

Table 4: Number of parameters and time per epoch for the models. For GMS-DH, the parameters in the decoder refers to both decoders.

|  | #Params Enc. | #Params Dec. | Epoch SG | Epoch MG |
|---|---|---|---|---|
| MBM | 2.38M | 266K | 4.7m | 4.7m |
| GMS-DH | 2.06M | 1.08M | 8.4m | 24m |
| GMS-EB | 1.98M | 266K | 51m | 1.7h |

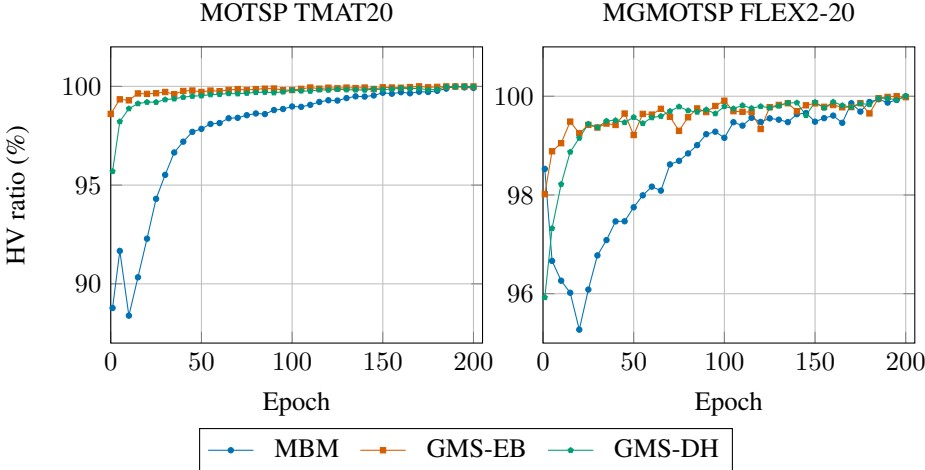

Figure 5: HV as percentage of best performance for models during training. GMS-EB and GMS-DH reach close to their final performance quickly and are more sample-efficient than MBM.

loss of the selection-decoder. That is, only the loss from the routing-decoder is utilized to calculate encoder gradients.

## D.2 TRAINING TIMES

Table 4 shows the number of parameters for the models as well as training time per epoch for one problem on a Simple Graph (SG) and Multigraph (MG). The SG problem is the MOTSP with 50 nodes, while the MG problem is the MGMOTSP with 50 nodes and 2 parallel edges. It can be noted that GMS-EB takes a very long time to train, especially on multigraphs, while GMS-DH is more efficient and MBM is the most efficient. In fact, MBM is not affected by the multigraph representation as it works on pruned graphs.

We want to highlight that the training performance is a weakness of GMS. However, these models are very sample efficient, which is illustrated on examples with 20 nodes (that is, without curriculum learning) in Figure 5. Moreover, we try to remedy this weakness through the curriculum approach. When utilizing the curriculum learning, it is often enough to train on the small instance sizes to obtain a majority of performance and the rest is obtained after a few epochs of training on the larger instance. This can be seen in Figure 6, which showcases two examples with 50 nodes.

## E ROUTING PROBLEMS AND DISTRIBUTIONS

In this section, we detail the distributions and routing problems which we solve.

## E.1 DISTRIBUTIONS

The following distributions for node distances are used across the problems on simple graphs.

- **EUC** - Euclidean. Here, we sample node coordinates uniformly and independently in the unit square. These are utilized to construct a distance matrix using the Euclidean distance.

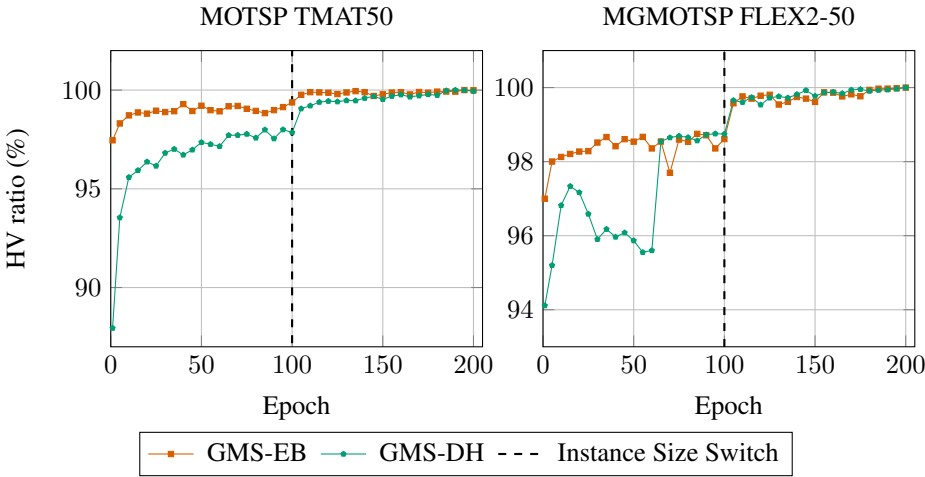

Figure 6: HV as percentage of best performance for GMS during training. It is enough to train on 20 nodes to obtain a majority ($\approx 98\%$) of the performance. After 5 epochs of training on instances with 50 nodes, the models reach $> 99.5\%$ of their final performance.

For a problem with $m$ node distances (e.g., the MOTSP with $m$ objectives) we sample $m$ pairs of coordinates independently.

- **TMAT** - Asymmetric distance matrix obeying the triangle inequality (Kwon et al., 2021). The distance matrix is first populated with random independent elements and then post-processed until the triangle inequality is satisfied. For a problem with $m$ distances, we sample $m$ matrices independently.

- **XASY** - Extremely asymmetric distance matrix (Gaile et al., 2022). The distance matrix does not satisfy the triangle inequality. It is obtained by independently sampling each element uniformly between 0 and 1. Again, we sample all $m$ matrices independently.

The following distributions are used for problems on multigraphs.

- **FLEX**$x$ - Flexible number of edges between each node pair. Here we sample $x$ vectors of dimension $m$ representing edge attributes for edges between a node pair. Each attribute for each edge is sampled uniformly between 0 and 1. Then we remove dominated edges until only a Pareto set of edges remains.

- **FIX**$x$ - Fixed number of edges between each node pair. We start by sampling $x$ vectors in the same way as the FLEX-distribution. Then we sort each attribute so that no edge is dominated. For $m = 2$, we sort one attribute in descending order and one attribute in ascending order. For $m = 3$ (showed in Appendix H), we sort one attribute in descending order, one in ascending order and keep one attribute in random order.

Note that neither FLEX$x$ nor FIX$x$ obeys the triangle inequality, although such an assumption holds for real instances. We highlight that these distributions are utilized since they are simple and easy to sample from, creating a more transparent picture of the model performance. Motivated by the fact that XASY seems more difficult to learn than TMAT, we hypothesize that the triangle inequality would contribute positively to model performance.

### E.2    MOTSP

**Problem Description.** In the multi-objective traveling salesman problem the goal is to visit all nodes/customers and return to the starting position. The Euclidean version of this problem has been widely treated in the learning-for-routing community, e.g., by Lin et al. (2022) and Fan et al. (2025a). In our case, we sample $m$ distances, corresponding to the $m$ objectives, between each pair of customers. The goal is to design a tour that minimizes the sum of the distances. A tour can start from any node.

**Data Generation**. Edge distances are sampled according to the EUC, TMAT and XASY distributions above.

**Model Details**. There are no node features for this problem, hence both the GREAT-based and MatNet-based encoders have the two distances as edge features, while the latter uses zero-vectors as "row" input.

**Evaluation Details**. For all distributions in the bi-objective case, we utilize $(15, 15)$, $(30, 30)$ and $(60, 60)$ as reference for HV calculation for 20, 50 and 100 nodes respectively. In the zero-shot experiments, we have $(90, 90)$ as reference for 150 nodes and $(120, 120)$ for 200 nodes. In the tri-objective case, we use $(15, 15, 15)$. $(30, 30, 30)$ and $(60, 60, 60)$ for 20, 50 and 100 nodes respectively.

### E.3 MOCVRP

**Problem Description**. In the multi-objective capacitated vehicle routing problem, a truck starts at a depot and must pick up goods from a specified number of customers. The vehicle has a fixed capacity while each customer has a demand. When the capacity is full, the vehicle must return to the depot. As many vehicles as required can be utilized to meet the demand of all customers.

Similarly to the TSP, this problem has been widely treated in the Euclidean setting using learning-based methods. As is common in the multi-objective setting (Lin et al., 2022), we assume one objective is the traveled distance, while the other is the longest tour for a single vehicle (makespan). As such, each node pair is only associated with one distance.

**Data Generation**. We assume that the capacity of the truck is 30, 40 and 50 for 20, 50 and 100 nodes respectively, while the demands are sampled uniformly from the set $\{1, \ldots, 9\}$. This is the same setting as in e.g., Kool et al. (2018) and Lin et al. (2022). Edge distances are sampled according to the EUC, TMAT and XASY distributions.

**Model Details**. In the decoding, we augment the routing context (represented by the query $q_t$) with the current load of the vehicle. For the GREAT-based encoders, we let the features of an edge be its distance as well as the demand of the customer at the end. On the other hand, the MatNet-based encoders only utilize the distance as an edge-feature while the demand is used as a node feature.

**Evaluation Details**. For all distributions, we utilize $(15, 3)$, $(40, 3)$ and $(60, 3)$ as reference for HV calculation for the three considered instance sizes.

### E.4 MGMOTSP

**Problem Description**. As for the MOTSP, the goal of the multigraph MOTSP is to visit all nodes and return to the start. The only difference is that the underlying structure is a multigraph. Similarly, we define $m$ distances for each node pair and these are associated with the objectives of the problem.

While this variant of the TSP on a multigraph has not been treated before to our knowledge (neither using learning nor other methods), some stochastic single-objective variants have been proposed by e.g., Tadei et al. (2017) and Maggioni et al. (2014). Moreover, we believe that the MGMOTSP is a suitable benchmark problem for our solvers, as it is simple and easy to find good baselines for.

**Data Generation**. We sample distances between the nodes according to the FLEX$x$ and FIX$x$ distributions.

**Model Details**. As the MOTSP, the MGMOTSP lacks node features. Hence, we utilize the same setup for the models as in the MOTSP. Moreover, for the MatNet-based model, we sparsify the underlying multigraph given each preference before it is fed into the encoder. This is done by removing all edges with suboptimal linearly scalarized cost compared to a parallel edge. To make this pruning strategy exact, we train MBM using linear reward instead of the Chebyshev reward.

**Evaluation Details**. For the FLEX$x$ distribution in the bi-objective case, we utilize $(15, 15)$, $(30, 30)$ and $(60, 60)$ as reference for instance sizes 20, 50 and 100, regardless of $x$. In the zero-shot experiments we set the references to $(90, 90)$ and $(120, 120)$ for FLEX$x$ with 150 and 200 nodes. For FIX$x$, the reference is given by $(N, N)$, where $N$ is the node count. When considering three objectives, we use $(15, 15, 15)$, $(30, 30, 30)$ and $(60, 60, 60)$ for FLEX$x$ and $(N, N, N)$ for FIX$x$.

### E.5 MGMOCVRP

**Problem Description**. Like in the MOCVRP, the goal in multigraph MOCVRP is to determine an optimal route serving all customers for a truck starting at a depot with fixed capacity. Unlike the MOCVRP described above, in the MGMOCVRP we do not consider the makespan as one objective and total distance as the other. Instead, we consider two conflicting total distances, giving rise to multiple conflicting edges between each node pair.

Scalarizing this problem and correspondingly pruning the multigraph ensures that the near-optimal baseline HGS (Vidal et al., 2012) is applicable to the scalarized problem. This ensures we can accurately assess model performance.

**Data Generation**. We sample distances between the nodes according to the FLEX$x$ and FIX$x$ distributions. Demands are sampled as in the MOCVRP.

**Model Details**. The same model setup is used as for the MOCVRP, with the exception that MBM uses linear pre-pruning of multigraphs into simple graphs (see MGMOTSP for details).

**Evaluation Details**. For the FLEX$x$ distribution we utilize $(15, 15)$, $(40, 40)$ and $(60, 60)$ as references with 20, 50 and 100 nodes. For FIX$x$, we always set the reference to $(N, N)$, where $N$ is the number of customers in the problem.

### E.6 MGMOTSPTW

**Problem Description**. The goal in MGMOTSPTW is to create a tour that visits all customers and returns to the starting position. However, in this case we assume there is a fixed depot and that each customer/node is associated with a time-window. Preferably, the vehicle should arrive within the time-window for each customer. As such, we let one objective be the number of violated time-windows while the other is the traveled distance. Accordingly, each edge connecting two locations is associated with a travel time and a distance.

Multi-objective problems with similar objectives are fairly widespread in literature (Legrand et al., 2025), although not on multigraphs. In the multigraph framework, this problem is reminiscent of the vehicle routing problem with time-windows of Ben Ticha et al. (2017), but with a second objective that is the soft variant of the time-window constraint.

**Data Generation**. Distances and travel times are sampled according to the FLEX$x$ and FIX$x$ distributions. Additionally, we sample time-windows according to the "medium" distribution of Chen et al. (2024) and Bi et al. (2024). In this setting, time-windows are sampled uniformly and independently, which results in difficult instances where there is a true trade-off between optimal paths and paths satisfying the time-window constraint. We also argue that this premise is simple for initial evaluation of our models, especially in comparison to e.g., the "hard" instances of Chen et al. (2024), which test out-of-distribution generalization.

Further, we assume that the vehicle can depart immediately from each customer, i.e., that there is no service time, and that no waiting occurs, even if a vehicle arrives before the starting time. The latter is for simplicity, as in general how long the vehicle should wait at a node is a decision in itself. We also argue that this assumption is fairly realistic.

**Model Details**. In the decoder, we enhance the routing context with an embedding of the current time. In the encoder, the node features in this case are the starting time and end time of the service, while the edge features are distance and travel time. Consequently, we augment the input features of GREAT so that an edge is associated with its distance, travel time and the time-window of the ending node. As usual for MatNet, the node features are used as "row" inputs while the edge features are used in the distance matrix.

Note that in this case, pruning of the multigraph for MatNet is not obvious. For instance, removing an edge with a high travel time but low distance might not be optimal if traversing that edge would result in no violated time-windows anyway. For simplicity, we apply the same simple pruning method as for the MGMOTSP. That is, we form the scalarized cost of each edge using a linear combination of its travel time and distance, then we remove all edges apart from the best one. We also choose the linear reward instead of the Chebyshev reward for MBM.

**Evaluation Details**. In this case, we let the reference value corresponding to the time-window violation be $N + 5$, while the reference value corresponding to the distance is the same as in the MGMOTSP case.

## F  SETUP FOR MOEAS

As a base for the implementation of NSGA-II, NSGA-III and MOEA/D, we utilized Pymoo[2] (Blank & Deb, 2020) and as a base for MOGLS, we utilized GeneticLocalSearch_TSP[3] (Li et al., 2021). For the former three, we limit the number of iterations to 1000 while the latter is limited to 10000. Additionally, local search strategies are a necessity for efficient usage of genetic algorithms in combinatorial optimization. For the MOTSP, we utilized the 2-opt heuristic of Jaszkiewicz (2002), while we utilized the problem-specific local search and chromosome decoding strategy of Lacomme et al. (2006) for the MOCVRP.

However, in both cases we find that by accepting the best local move instead of just an improving one, the performance on large asymmetric problems improves substantially. As this affects the runtime negatively (due to the need to search through all moves), we limit the number of local moves in each search depending on the problem size. Additionally, for the algorithms not requiring preference weights for each individual (NSGA-II and NSGA-III), we must calculate such weights to compare the improvements. In the bi-objective case, this is done using the strategy of Lacomme et al. (2006) according to

$$
\begin{aligned}
\widetilde{w}_1(S) &= (f_1(S) - f_1^{\min})/(f_1^{\max} - f_1^{\min}), \\
\widetilde{w}_2(S) &= (f_2(S) - f_2^{\min})/(f_2^{\max} - f_2^{\min}), \\
w_1(S) &= \widetilde{w}_1(S)/(\widetilde{w}_1(S) + \widetilde{w}_2(S)), \\
w_2(S) &= \widetilde{w}_2(S)/(\widetilde{w}_1(S) + \widetilde{w}_2(S)),
\end{aligned}
\tag{26}
$$

where $S$ is the solution being improved and $f_i^{\min}, f_i^{\max}, i = 1, 2$ are minimum and maximum values for the respective objectives.

In the multigraph setting, we design our own setup to tailor the MOEAs. To start with, inspired by one representation from Beke et al. (2021) regarding shortest path problems, we let a chromosome for a tour be given by a permutation of the nodes multiplied with 100 added with the index of the edge to the next node. For example, in an MGMOTSP with 2 parallel edges and 5 nodes, a chromosome might be

$$(101, 302, 201, 501, 402).$$

For the mutation operator, we randomly either change one edge or reverse a segment of the tour. For the crossover, we utilize an edge recombination operator. Regarding the local search, we use a variant of 2-opt where we again accept the best local move. A local move is defined by a standard 2-opt move with the nodes, together with new edge selections given the changed node order. This edge selection is found by choosing the edge with the lowest linearly scalarized cost between nodes that were previously not neighbors or have changed order.

We want to stress that the focus in this study is not to design evolutionary algorithms, we simply want baselines to compare against. We find that the proposed approach is somewhat efficient for small node counts, but performs quite badly for large graphs.

## G  ABLATION STUDY

Here, we present additional results obtained by replacing the selection-decoder in GMS-DH with a simple pruning function that discards edges with suboptimal linearly scalarized cost. As shown in Section 5.3, this approach performs rather poorly on the MGMOTSPTW. However, it is more reasonable to assume that it will yield good results for the MGMOTSP.

To ensure a fair comparison, we evaluate both learned pruning and simple pruning using linearly scalarized rewards and Chebyshev rewards. In the linear case, simple pruning is optimal by Proposition 1. Moreover, the learned pruning model is trained using $K_1 = 4$ edge selections per instance.

---

[2]https://pymoo.org/
[3]https://github.com/kevin031060/Genetic_Local_Search_TSP

Table 5: Ablation of the learned pruning in vanilla GMS-DH by replacing it with a simple pruning function. Results are for the MGMOTSP. The performance of GMS-DH PP from Table 1 is repeated for convenience.

| | FLEX2-50 | | FIX2-50 | |
|---|---|---|---|---|
| | HV | Gap | HV | Gap |
| GMS-DH Learned (Ch. Reward) | **0.89** | **1.84%** | 0.91 | 1.73% |
| GMS-DH Learned (Lin. Reward) | 0.88 | 2.74% | **0.91** | **1.66%** |
| GMS-DH Simple (Ch. Reward) | 0.88 | 2.00% | 0.90 | 1.82% |
| GMS-DH Simple (Lin. Reward) | 0.88 | 2.65% | 0.91 | 1.76% |
| GMS-DH PP (Lin. Reward) | 0.77 | 14.24% | 0.82 | 10.51% |

Accordingly, we use $4\times$ more POMO samples when training the model with simple pruning to match the training budget.

Ablation results are shown in Table 5. For the MGMOTSP, simple and learned pruning perform quite similarly (although learned pruning shows a slight advantage under Chebyshev rewards for both distributions). Indeed, these results illustrate that it can be a smart approach to replace the selection-decoder in GMS-DH if a sound heuristic or exact pruning method is available. However, pruning after the encoder clearly outperforms pre-pruning (as used in GMS-DH PP), both in terms of solution quality and runtime, since pre-pruning requires reencoding the graph for each preference.

## H COMPLEMENTARY NUMERICAL RESULTS

In this section, we show additional numerical results for more problems, distributions, benchmarks and instance sizes.

### H.1 ADDITIONAL NEURAL BASELINES

Here, we show more results for naive extensions of single-objective neural solvers for asymmetric problems, applied to multi-objective asymmetric problems and multigraph problems. More specifically, we implement a GREAT-Based Model (GRBM) and a GOAL-Based Model (GOBM), utilizing GREAT NB layers (Lischka et al., 2025) and GOAL mixed-attention layers (Drakulic et al., 2025) in the encoder. The rest of the framework is the same as for MBM. That is, we utilize the POMO framework for training and preference-conditioned multi-pointer networks for decoding. In the multigraph scenarios, we also pre-prune the graphs to handle edge selection and to ensure the the input-structure is compatible for GOBM.

The corresponding results are displayed in Table 6. While GRBM has a slightly lower gap compared to GMS-DH on TMAT50 and XASY50 for the MOCVRP, and GOBM has a slightly lower gap on FIX2-100 for the MGMOTSP, both GRBM and GOBM fail to perform robustly across distributions, graph sizes and problems. In contrast, GMS-DH and GMS-EB maintain a strong performance irrespective of the graph distribution and routing problem. GMS-DH performs better than GRBM and GOBM in 13 of the 16 cases displayed in Table 6, often quite significantly.

### H.2 PROBLEMS ON SIMPLE GRAPHS

Table 7 and 8 show results for the MOTSP and MOCVRP respectively. Apart from the benchmarks in the main text, we present the performance of three simple greedy heuristics on the MOTSP: Nearest Neighbor (NN), Nearest Insertion (NI) and Farthest Insertion (FI). Like LKH and OR-tools, these utilize the linear scalarization to transform the MO problem into single-objective subproblems. Additionally, we show results on Euclidean instances (EUC). For these, there are many existing models in literature and thus we replace MBM with the Conditional Neural Heuristic (CNH) of Fan et al. (2025a). This model is augmented with the Euclidean augmentation of Lin et al. (2022).

On the EUC distribution, our models are generally slightly worse than CNH. This is to be expected though, as a common pattern in previous studies is that architectures built for directed graph inputs perform worse on EUC than those specialized for coordinate inputs (Lischka et al., 2025; Kwon et al., 2021).

Table 6: Additional neural baselines for the MOTSP, MOCVRP, MGMOTSP and MGMOCVRP across graph sizes (50 and 100 nodes) and distributions. We also include GMS-DH for easy performance comparisons. Note that gap is with respect to the best performing method in Table 1.

| | | TMAT50 | | | TMAT100 | | | XASY50 | | | XASY100 | | |
|---|---|---|---|---|---|---|---|---|---|---|---|---|---|
| | | HV | Gap | Time | HV | Gap | Time | HV | Gap | Time | HV | Gap | Time |
| **MOTSP** | GOBM | 0.49 | 14.92% | (3.6s) | 0.55 | 13.95% | (19s) | 0.77 | 7.35% | (3.6s) | 0.83 | 7.88% | (19s) |
| | GOBM (aug) | 0.50 | 13.06% | (33s) | 0.55 | 12.91% | (2.7m) | 0.79 | 5.16% | (33s) | 0.84 | 6.56% | (2.7m) |
| | GRBM | 0.50 | 14.53% | (5.5s) | 0.56 | 11.53% | (20s) | 0.80 | 4.20% | (5.6s) | 0.84 | 6.54% | (20s) |
| | GRBM (aug) | 0.50 | 14.06% | (33s) | 0.56 | 11.13% | (2.9m) | 0.81 | 3.02% | (33s) | 0.85 | 6.05% | (2.9m) |
| | GMS-DH | 0.57 | 2.36% | (4.1s) | 0.62 | 2.76% | (20s) | 0.80 | 4.12% | (3.9s) | 0.86 | 4.87% | (20s) |
| | GMS-DH (aug) | 0.57 | 1.76% | (28s) | 0.62 | 2.19% | (2.6m) | 0.81 | 2.84% | (28s) | 0.87 | 3.85% | (2.6m) |
| | | TMAT50 | | | TMAT100 | | | XASY50 | | | XASY100 | | |
| **MOCVRP** | GOBM | 0.45 | 5.60% | (6.2s) | 0.42 | 6.58% | (28s) | 0.67 | 11.22% | (6.2s) | 0.66 | 16.67% | (28s) |
| | GOBM (aug) | 0.45 | 4.29% | (40s) | 0.43 | 4.94% | (3.5m) | 0.70 | 7.23% | (40s) | 0.70 | 12.48% | (3.5m) |
| | GRBM | 0.47 | 0.80% | (10s) | 0.37 | 18.79% | (31s) | 0.73 | 3.40% | (11s) | 0.77 | 3.64% | (30s) |
| | GRBM (aug) | 0.47 | 0.08% | (46s) | 0.38 | 15.63% | (3.9m) | 0.74 | 1.20% | (45s) | 0.78 | 1.52% | (3.9m) |
| | GMS-DH | 0.47 | 0.76% | (6.8s) | 0.45 | 1.46% | (32s) | 0.72 | 4.13% | (6.6s) | 0.77 | 3.60% | (33s) |
| | GMS-DH (aug) | 0.47 | 0.36% | (47s) | 0.45 | 0.79% | (4.3m) | 0.74 | 1.65% | (47s) | 0.79 | 1.21% | (4.7m) |
| | | FLEX2-50 | | | FLEX2-100 | | | FIX2-50 | | | FIX2-100 | | |
| **MGMOTSP** | GOBM | 0.86 | 4.91% | (12s) | 0.91 | 3.97% | (46s) | 0.89 | 3.02% | (12s) | 0.93 | 2.04% | (46s) |
| | GOBM (aug) | 0.87 | 3.47% | (1.4m) | 0.92 | 3.08% | (6.1m) | 0.90 | 2.13% | (1.4m) | 0.94 | 1.52% | (6.2m) |
| | GRBM | 0.77 | 14.53% | (40s) | 0.83 | 11.96% | (2.7m) | 0.89 | 3.27% | (40s) | 0.93 | 2.42% | (2.7m) |
| | GRBM (aug) | 0.79 | 12.77% | (5.1m) | 0.85 | 9.81% | (21m) | 0.90 | 2.90% | (5.1m) | 0.93 | 2.24% | (21m) |
| | GMS-DH | 0.89 | 1.84% | (13s) | 0.92 | 2.11% | (49s) | 0.91 | 1.73% | (12s) | 0.93 | 2.13% | (52s) |
| | GMS-DH (aug) | 0.89 | 1.45% | (1.5m) | 0.93 | 1.61% | (6.6m) | 0.91 | 1.40% | (1.5m) | 0.93 | 1.92% | (6.6m) |
| | | FLEX2-50 | | | FLEX2-100 | | | FIX2-50 | | | FIX2-100 | | |
| **MGMOCVRP** | GOBM | 0.83 | 7.02% | (13s) | 0.82 | 8.18% | (55s) | 0.83 | 5.52% | (13s) | 0.86 | 5.60% | (55s) |
| | GOBM (aug) | 0.84 | 5.36% | (1.6m) | 0.83 | 6.66% | (7.1m) | 0.84 | 4.15% | (1.6m) | 0.87 | 4.58% | (7.1m) |
| | GRBM | 0.82 | 7.88% | (46s) | 0.80 | 9.51% | (2.9m) | 0.76 | 13.11% | (46s) | 0.80 | 11.91% | (2.9m) |
| | GRBM (aug) | 0.83 | 6.54% | (5.7m) | 0.82 | 8.04% | (23m) | 0.80 | 8.23% | (5.6m) | 0.85 | 6.68% | (23m) |
| | GMS-DH | 0.85 | 4.86% | (20s) | 0.85 | 4.94% | (1.0m) | 0.83 | 4.96% | (20s) | 0.88 | 3.58% | (1.0m) |
| | GMS-DH (aug) | 0.85 | 3.86% | (2.1m) | 0.85 | 3.89% | (8.2m) | 0.84 | 3.96% | (2.1m) | 0.88 | 3.05% | (8.1m) |

### H.3 MULTIGRAPH PROBLEMS

We display additional results for the MGMOTSP, MGMOCVRP and MGMOTSPTW in Tables 9-12. As for the MOTSP, we include NN, NI and FI for MGMOTSP and NN for the MGMOCVRP. In the case of MGMOTSPTW, we also include results for LKH and OR-tools when treating the problem as the MGMOTSP. That is, we neglect the time-windows and treat the time duration of each edge exactly as the distance. We call these methods implicit LKH and implicit OR-tools, as they implicitly minimize the time-window violations by reducing the time the tour takes. Additionally, for the MGMOTSP, we show results for FLEX5 and FIX5, i.e., when there are 5 parallel edges between each node pair.

### H.4 GENERALIZATION TO LARGER INSTANCES

In Tables 13-15, we report more generalization results for our methods in the MOTSP and MG-MOTSP cases. Notably, even when introducing significantly more edges by testing on FLEX10 and FIX10, our methods remain competitive zero-shot. The performance of GMS-DH degrades somewhat for FIX10-100, which we attribute to the growing action space of the selection-decoder.

### H.5 TRI-OBJECTIVE PROBLEMS

Finally, we show results when considering three objectives, rather than two as in the previous cases. In this case, we restrict ourselves to the MOTSP and MGMOTSP problems. We display the results in Tables 16 and 17. Generally, our methods seem to scale well with the number of objectives. The exception is GMS-DH on the XASY distribution for the MOTSP with 100 nodes, which seems to underperform compared to, for example, MBM and OR tools. Otherwise, both GMS-EB and GMS-DH show competitive results with a comparably short inference time. As expected, CNH performs slightly better on the Euclidean distribution compared to our methods.

Table 7: MOTSP with different instance sizes (20, 50 and 100) and distributions.

| | EUC20 | | | EUC50 | | | EUC100 | | |
| --- | --- | --- | --- | --- | --- | --- | --- | --- | --- |
| | HV | Gap | Time | HV | Gap | Time | HV | Gap | Time |
| LKH | 0.52 | 0.00% | (2.6m) | 0.58 | 0.00% | (14m) | 0.68 | 0.00% | (1.2h) |
| OR Tools | 0.51 | 0.97% | (2.0m) | 0.57 | 1.83% | (15m) | 0.67 | 1.57% | (1.3h) |
| NN | 0.46 | 11.44% | (1.9s) | 0.51 | 11.42% | (7.1s) | 0.63 | 8.23% | (25s) |
| NI | 0.47 | 8.53% | (11s) | 0.52 | 9.97% | (1.8m) | 0.63 | 7.82% | (12m) |
| FI | 0.50 | 3.44% | (13s) | 0.55 | 5.79% | (2.1m) | 0.65 | 4.96% | (14m) |
| MOGLS | **0.51** | **0.79%** | (1.0h) | **0.58** | **0.74%** | (3.8h) | **0.68** | **1.16%** | (32h) |
| NSGA-II | 0.50 | 2.63% | (44m) | 0.52 | 11.18% | (3.4h) | 0.54 | 21.13% | (12h) |
| NSGA-III | 0.50 | 3.56% | (46m) | 0.50 | 14.07% | (3.4h) | 0.53 | 21.82% | (12h) |
| MOEA/D | 0.51 | 0.87% | (56h) | 0.57 | 2.53% | (3.6h) | 0.63 | 8.08% | (13h) |
| CNH | 0.51 | 1.16% | (6.5s) | 0.57 | 1.84% | (7.1s) | 0.67 | 1.89% | (25s) |
| CNH (aug) | 0.51 | 0.85% | (17s) | 0.57 | 1.26% | (44s) | 0.67 | 1.45% | (3.1m) |
| GMS-EB | 0.51 | 0.93% | (2.0s) | 0.57 | 1.98% | (25s) | 0.67 | 2.34% | (3.6m) |
| GMS-EB (aug) | 0.51 | 0.81% | (14s) | 0.57 | 1.57% | (3.3m) | 0.67 | 2.09% | (28m) |
| GMS-DH | 0.51 | 1.30% | (1.1s) | 0.57 | 2.64% | (3.8s) | 0.67 | 2.49% | (20s) |
| GMS-DH (aug) | 0.51 | 0.93% | (5.3s) | 0.57 | 1.91% | (27s) | 0.67 | 2.24% | (2.6m) |

| | TMAT20 | | | TMAT50 | | | TMAT100 | | |
| --- | --- | --- | --- | --- | --- | --- | --- | --- | --- |
| LKH | 0.58 | 0.00% | (2.3m) | 0.58 | 0.00% | (6.4m) | 0.63 | 0.00% | (29m) |
| OR Tools | 0.56 | 3.01% | (3.3m) | 0.54 | 6.13% | (29m) | 0.59 | 7.12% | (2.5h) |
| NN | 0.48 | 16.66% | (1.9s) | 0.47 | 18.34% | (7.0s) | 0.54 | 15.25% | (25s) |
| NI | 0.53 | 8.78% | (11s) | 0.50 | 13.17% | (1.8m) | 0.55 | 13.24% | (12m) |
| FI | 0.54 | 6.60% | (13s) | 0.52 | 10.65% | (2.0m) | 0.57 | 10.84% | (14m) |
| MOGLS | 0.54 | 7.72% | (20m) | 0.40 | 30.90% | (2.7h) | 0.38 | 40.03% | (19h) |
| NSGA-II | 0.57 | 2.17% | (44m) | 0.51 | 12.91% | (3.5h) | 0.47 | 25.98% | (12h) |
| NSGA-III | 0.57 | 2.87% | (45m) | 0.50 | 14.54% | (3.5h) | 0.46 | 27.53% | (13h) |
| MOEA/D | 0.57 | 1.43% | (56m) | 0.53 | 9.51% | (3.7h) | 0.50 | 20.93% | (13h) |
| MBM | 0.57 | 2.29% | (1.3s) | 0.50 | 13.51% | (3.9s) | 0.56 | 11.16% | (19s) |
| MBM (aug) | 0.58 | 0.96% | (5.6s) | 0.52 | 10.17% | (27s) | 0.58 | 9.17% | (2.4m) |
| GMS-EB | 0.58 | 0.88% | (2.2s) | 0.57 | 1.50% | (25s) | 0.63 | 1.45% | (3.6m) |
| GMS-EB (aug) | **0.58** | **0.67%** | (15s) | **0.57** | **1.14%** | (3.3m) | **0.63** | **1.13%** | (29m) |
| GMS-DH | 0.58 | 1.08% | (1.3s) | 0.57 | 2.36% | (4.1s) | 0.62 | 2.76% | (20s) |
| GMS-DH (aug) | 0.58 | 0.79% | (5.6s) | 0.57 | 1.76% | (28s) | 0.62 | 2.19% | (2.6m) |

| | XASY20 | | | XASY50 | | | XASY100 | | |
| --- | --- | --- | --- | --- | --- | --- | --- | --- | --- |
| LKH | 0.75 | 0.00% | (2.3m) | 0.83 | 0.00% | (6.9m) | 0.90 | 0.00% | (32m) |
| OR Tools | 0.71 | 5.49% | (3.0m) | 0.77 | 7.25% | (24m) | 0.85 | 6.06% | (1.8h) |
| NN | 0.60 | 20.11% | (1.8s) | 0.70 | 16.32% | (7.0s) | 0.80 | 10.94% | (25s) |
| NI | 0.61 | 17.76% | (11s) | 0.66 | 21.37% | (1.8m) | 0.73 | 18.88% | (12m) |
| FI | 0.60 | 19.57% | (12s) | 0.64 | 23.62% | (2.1m) | 0.72 | 20.42% | (14m) |
| MOGLS | 0.65 | 13.23% | (19m) | 0.51 | 39.16% | (2.6h) | 0.47 | 47.82% | (19h) |
| NSGA-II | 0.72 | 4.06% | (44m) | 0.70 | 15.63% | (3.5h) | 0.64 | 29.74% | (13h) |
| NSGA-III | 0.71 | 4.56% | (45m) | 0.70 | 16.64% | (3.5h) | 0.63 | 30.13% | (13h) |
| MOEA/D | 0.73 | 2.68% | (56m) | 0.73 | 12.68% | (3.6h) | 0.70 | 22.22% | (13h) |
| MBM | 0.68 | 9.10% | (1.4s) | 0.75 | 10.37% | (3.7s) | 0.84 | 7.41% | (19s) |
| MBM (aug) | 0.70 | 5.86% | (5.5s) | 0.76 | 8.28% | (27s) | 0.85 | 6.34% | (2.4m) |
| GMS-EB | 0.73 | 1.95% | (2.3s) | 0.81 | 2.94% | (25s) | 0.88 | 3.03% | (3.6m) |
| GMS-EB (aug) | **0.74** | **1.30%** | (15s) | **0.82** | **2.06%** | (3.3m) | **0.88** | **2.76%** | (28m) |
| GMS-DH | 0.73 | 2.37% | (1.9s) | 0.80 | 4.12% | (3.9s) | 0.86 | 4.87% | (20s) |
| GMS-DH (aug) | 0.74 | 1.57% | (5.8s) | 0.81 | 2.84% | (28s) | 0.87 | 3.85% | (2.6m) |

Table 8: MOCVRP with different instance sizes (20, 50 and 100) and distributions.

| | EUC20 | | | EUC50 | | | EUC100 | | |
| | HV | Gap | Time | HV | Gap | Time | HV | Gap | Time |
|---|---|---|---|---|---|---|---|---|---|
| MOGLS | 0.23 | 3.35% | (4.1h) | 0.17 | 37.39% | (26h) | 0.00 | 100.00% | (108h) |
| NSGA-II | 0.23 | 0.13% | (2.6h) | 0.27 | 2.14% | (24h) | 0.20 | 18.91% | (75h) |
| NSGA-III | 0.23 | 0.21% | (2.7h) | 0.26 | 2.91% | (25h) | 0.20 | 21.23% | (75h) |
| MOEA/D | 0.23 | 0.90% | (2.9h) | 0.26 | 2.73% | (25h) | 0.23 | 7.40% | (73h) |
| CNH | 0.23 | 0.77% | (5.0s) | 0.27 | 0.59% | (9.5s) | 0.25 | 0.56% | (29s) |
| CNH (aug) | **0.23** | **0.00%** | (29s) | **0.27** | **0.00%** | (58s) | **0.25** | **0.00%** | (3.6m) |
| GMS-EB | 0.23 | 0.30% | (3.4s) | 0.27 | 0.44% | (33s) | 0.25 | 0.56% | (4.1m) |
| GMS-EB (aug) | 0.23 | 0.13% | (22s) | 0.27 | 0.18% | (4.7m) | 0.25 | 0.32% | (54m) |
| GMS-DH | 0.23 | 1.33% | (2.3s) | 0.27 | 1.11% | (6.6s) | 0.25 | 1.36% | (30s) |
| GMS-DH (aug) | 0.23 | 0.90% | (10s) | 0.27 | 0.66% | (46s) | 0.25 | 0.92% | (4.1m) |

| | TMAT20 | | | TMAT50 | | | TMAT100 | | |
| | HV | Gap | Time | HV | Gap | Time | HV | Gap | Time |
|---|---|---|---|---|---|---|---|---|---|
| MOGLS | 0.40 | 1.78% | (4.1h) | 0.31 | 35.06% | (28h) | 0.05 | 89.38% | (116h) |
| NSGA-II | 0.40 | 1.38% | (2.7h) | 0.36 | 23.04% | (23h) | 0.13 | 71.72% | (76h) |
| NSGA-III | 0.40 | 1.98% | (2.7h) | 0.35 | 25.91% | (24h) | 0.12 | 74.11% | (76h) |
| MOEA/D | 0.39 | 2.59% | (3.0h) | 0.41 | 13.98% | (24h) | 0.19 | 58.70% | (76h) |
| MBM | 0.40 | 1.14% | (2.4s) | 0.47 | 1.33% | (6.2s) | 0.43 | 4.50% | (27s) |
| MBM (aug) | 0.40 | 0.86% | (10s) | 0.47 | 0.99% | (41s) | 0.44 | 3.29% | (3.3m) |
| GMS-EB | 0.40 | 0.25% | (3.5s) | 0.47 | 0.25% | (36s) | 0.45 | 0.57% | (4.2m) |
| GMS-EB (aug) | **0.41** | **0.00%** | (23s) | **0.47** | **0.00%** | (5.2m) | **0.45** | **0.00%** | (33m) |
| GMS-DH | 0.40 | 0.64% | (2.5s) | 0.47 | 0.76% | (6.8s) | 0.45 | 1.46% | (32s) |
| GMS-DH (aug) | 0.40 | 0.32% | (10s) | 0.47 | 0.36% | (47s) | 0.45 | 0.79% | (4.3m) |

| | XASY20 | | | XASY50 | | | XASY100 | | |
| | HV | Gap | Time | HV | Gap | Time | HV | Gap | Time |
|---|---|---|---|---|---|---|---|---|---|
| MOGLS | 0.59 | 2.39% | (4.0h) | 0.49 | 34.85% | (29h) | 0.13 | 83.50% | (121h) |
| NSGA-II | 0.60 | 1.24% | (2.7h) | 0.55 | 27.19% | (24h) | 0.28 | 64.39% | (85h) |
| NSGA-III | 0.59 | 2.21% | (2.7h) | 0.52 | 31.06% | (25h) | 0.25 | 68.40% | (86h) |
| MOEA/D | 0.59 | 3.24% | (3.0h) | 0.65 | 13.88% | (24h) | 0.42 | 47.23% | (83h) |
| MBM | 0.57 | 5.95% | (2.3s) | 0.70 | 6.57% | (6.2s) | 0.60 | 24.47% | (27s) |
| MBM (aug) | 0.59 | 3.06% | (10s) | 0.72 | 4.13% | (42s) | 0.64 | 19.60% | (3.4m) |
| GMS-EB | 0.60 | 1.44% | (3.5s) | 0.74 | 1.89% | (36s) | 0.79 | 1.23% | (4.3m) |
| GMS-EB (aug) | **0.61** | **0.00%** | (23s) | **0.75** | **0.00%** | (5.0m) | **0.80** | **0.00%** | (48m) |
| GMS-DH | 0.58 | 3.40% | (2.4s) | 0.72 | 4.13% | (6.6s) | 0.77 | 3.60% | (33s) |
| GMS-DH (aug) | 0.60 | 1.29% | (10s) | 0.74 | 1.65% | (47s) | 0.79 | 1.21% | (4.7m) |

Table 9: MGMOTSP on FLEX$x$ with different instances sizes (20, 50 and 100) and number of edges (2, 5).

| | FLEX2-20 | | | FLEX2-50 | | | FLEX2-100 | | |
| --- | --- | --- | --- | --- | --- | --- | --- | --- | --- |
| | HV | Gap | Time | HV | Gap | Time | HV | Gap | Time |
| LKH | 0.85 | 0.00% | (1.2m) | 0.90 | 0.00% | (6.3m) | 0.94 | 0.00% | (31m) |
| OR Tools | 0.82 | 3.30% | (3.0m) | 0.86 | 4.21% | (24m) | 0.91 | 3.46% | (1.8h) |
| NN | 0.74 | 13.34% | (1.9s) | 0.81 | 10.38% | (7.1s) | 0.88 | 6.71% | (25s) |
| NI | 0.75 | 11.13% | (9.6s) | 0.78 | 13.13% | (1.7m) | 0.84 | 11.32% | (12m) |
| FI | 0.75 | 11.98% | (12s) | 0.77 | 14.30% | (2.0m) | 0.83 | 12.20% | (14m) |
| MOGLS | 0.71 | 16.29% | (1.9h) | 0.49 | 45.84% | (29h) | 0.34 | 64.38% | (129h) |
| NSGA-II | 0.81 | 5.06% | (1.3h) | 0.74 | 17.62% | (19h) | 0.64 | 31.98% | (103h) |
| NSGA-III | 0.80 | 5.82% | (1.3h) | 0.74 | 18.48% | (19h) | 0.63 | 33.01% | (104h) |
| MOEA/D | 0.80 | 6.02% | (1.4h) | 0.74 | 17.50% | (18h) | 0.68 | 28.46% | (99h) |
| MBM | 0.82 | 3.30% | (4.4s) | 0.86 | 5.21% | (13s) | 0.91 | 3.72% | (44s) |
| MBM (aug) | 0.84 | 1.50% | (32s) | 0.87 | 3.98% | (1.7m) | 0.91 | 3.28% | (5.9m) |
| GMS-EB | 0.83 | 1.61% | (4.0s) | 0.88 | 2.00% | (56s) | 0.93 | 1.81% | (9.1m) |
| GMS-EB (aug) | 0.84 | 0.95% | (34s) | 0.89 | 1.50% | (7.5m) | **0.93** | **1.47%** | (1.2h) |
| GMS-DH | 0.84 | 1.34% | (3.2s) | 0.89 | 1.84% | (13s) | 0.92 | 2.11% | (49s) |
| GMS-DH (aug) | **0.84** | **0.86%** | (19s) | **0.89** | **1.45%** | (1.5m) | 0.93 | 1.61% | (6.6m) |

| | FLEX5-20 | | | FLEX5-50 | | | FLEX5-100 | | |
| --- | --- | --- | --- | --- | --- | --- | --- | --- | --- |
| LKH | 0.93 | 0.00% | (1.3m) | 0.95 | 0.00% | (6.4m) | 0.97 | 0.00% | (30m) |
| OR Tools | 0.91 | 1.51% | (2.9m) | 0.93 | 2.01% | (25m) | 0.96 | 1.60% | (1.8h) |
| NN | 0.86 | 6.78% | (1.8s) | 0.90 | 5.24% | (7.2s) | 0.94 | 3.33% | (28s) |
| NI | 0.88 | 5.48% | (9.5s) | 0.89 | 6.41% | (1.7m) | 0.92 | 5.45% | (12m) |
| FI | 0.87 | 5.89% | (11s) | 0.89 | 7.03% | (2.0m) | 0.92 | 5.89% | (14m) |
| MOGLS | 0.84 | 9.71% | (2.4h) | 0.69 | 27.67% | (52h) | 0.57 | 41.21% | (222h) |
| NSGA-II | 0.88 | 5.19% | (1.4h) | 0.80 | 16.61% | (31h) | 0.71 | 26.79% | (215h) |
| NSGA-III | 0.87 | 5.85% | (1.4h) | 0.79 | 17.48% | (31h) | 0.70 | 27.98% | (215h) |
| MOEA/D | 0.88 | 5.07% | (1.5h) | 0.82 | 13.49% | (30h) | 0.76 | 22.05% | (210h) |
| MBM | 0.90 | 2.38% | (4.4s) | 0.93 | 2.48% | (13s) | 0.96 | 1.71% | (45s) |
| MBM (aug) | 0.92 | 1.26% | (32s) | 0.93 | 2.12% | (1.7m) | 0.96 | 1.44% | (5.9m) |
| GMS-EB | 0.92 | 0.87% | (8.1s) | 0.94 | 1.08% | (2.3m) | 0.96 | 1.11% | (23m) |
| GMS-EB (aug) | **0.92** | **0.59%** | (1.1m) | **0.95** | **0.86%** | (18m) | **0.96** | **0.96%** | (3.0h) |
| GMS-DH | 0.92 | 1.03% | (3.5s) | 0.94 | 1.64% | (16s) | 0.96 | 1.66% | (1.3m) |
| GMS-DH (aug) | 0.92 | 0.64% | (23s) | 0.94 | 1.31% | (2.2m) | 0.96 | 1.45% | (9.9m) |

Table 10: MGMOTSP on FIX$x$ with different instances sizes (20, 50 and 100) and number of edges (2, 5).

| | FIX2-20 | | | FIX2-50 | | | FIX2-100 | | |
|---|---|---|---|---|---|---|---|---|---|
| | HV | Gap | Time | HV | Gap | Time | HV | Gap | Time |
| LKH | 0.85 | 0.00% | (1.2m) | 0.92 | 0.00% | (6.3m) | 0.95 | 0.00% | (30m) |
| OR Tools | 0.83 | 2.42% | (2.9m) | 0.90 | 2.59% | (24m) | 0.93 | 2.25% | (2.0h) |
| NN | 0.77 | 9.86% | (1.9s) | 0.86 | 6.34% | (7.1s) | 0.91 | 4.23% | (25s) |
| NI | 0.78 | 8.39% | (9.6s) | 0.85 | 7.95% | (1.7m) | 0.88 | 7.14% | (12m) |
| FI | 0.78 | 9.23% | (11s) | 0.84 | 8.82% | (2.0m) | 0.88 | 7.89% | (14m) |
| MOGLS | 0.74 | 13.68% | (1.9h) | 0.65 | 29.64% | (29h) | 0.54 | 42.73% | (122h) |
| NSGA-II | 0.81 | 5.64% | (1.3h) | 0.77 | 16.56% | (19h) | 0.69 | 27.86% | (96h) |
| NSGA-III | 0.80 | 6.85% | (1.3h) | 0.75 | 18.79% | (19h) | 0.66 | 30.17% | (95h) |
| MOEA/D | 0.80 | 6.20% | (1.4h) | 0.78 | 15.42% | (18h) | 0.72 | 24.69% | (91h) |
| MBM | 0.84 | 1.57% | (4.4s) | 0.90 | 2.46% | (13s) | 0.93 | 2.41% | (44s) |
| MBM (aug) | **0.85** | **0.50%** | (32s) | 0.91 | 1.66% | (1.7m) | 0.93 | 2.10% | (5.9m) |
| GMS-EB | 0.84 | 1.46% | (3.9s) | 0.91 | 1.27% | (57s) | 0.94 | 1.33% | (9.2m) |
| GMS-EB (aug) | 0.85 | 0.99% | (28s) | **0.91** | **1.02%** | (7.5m) | **0.94** | **1.09%** | (1.2h) |
| GMS-DH | 0.84 | 1.37% | (3.2s) | 0.91 | 1.73% | (12s) | 0.93 | 2.13% | (52s) |
| GMS-DH (aug) | 0.85 | 0.82% | (18s) | 0.91 | 1.40% | (1.5m) | 0.93 | 1.92% | (6.6m) |
| | FIX5-20 | | | FIX5-50 | | | FIX5-100 | | |
| LKH | 0.84 | 0.00% | (56s) | 0.89 | 0.00% | (6.2m) | 0.91 | 0.00% | (29m) |
| OR Tools | 0.82 | 1.41% | (3.0m) | 0.87 | 1.78% | (25m) | 0.90 | 1.78% | (2.0h) |
| NN | 0.79 | 5.97% | (1.9s) | 0.85 | 4.22% | (7.2s) | 0.88 | 3.11% | (25s) |
| NI | 0.79 | 4.98% | (9.8s) | 0.84 | 5.27% | (1.7m) | 0.86 | 5.24% | (12m) |
| FI | 0.79 | 6.00% | (11s) | 0.83 | 6.45% | (2.0m) | 0.85 | 6.37% | (14m) |
| MOGLS | 0.74 | 11.00% | (2.5h) | 0.67 | 24.51% | (38h) | 0.62 | 32.11% | (253h) |
| NSGA-II | 0.79 | 5.96% | (1.5h) | 0.72 | 18.65% | (32h) | 0.63 | 30.45% | (216h) |
| NSGA-III | 0.77 | 7.78% | (1.5h) | 0.69 | 21.60% | (32h) | 0.60 | 33.77% | (218h) |
| MOEA/D | 0.79 | 5.44% | (1.5h) | 0.77 | 13.45% | (31h) | 0.71 | 22.40% | (230h) |
| MBM | 0.82 | 1.45% | (4.4s) | 0.87 | 2.34% | (13s) | 0.89 | 1.94% | (44s) |
| MBM (aug) | **0.83** | **0.47%** | (32s) | 0.87 | 1.82% | (1.7m) | 0.90 | 1.69% | (5.9m) |
| GMS-EB | 0.83 | 1.05% | (8.1s) | 0.88 | 1.03% | (2.3m) | 0.90 | 1.08% | (23m) |
| GMS-EB (aug) | 0.83 | 0.77% | (1.1m) | **0.88** | **0.89%** | (19m) | **0.90** | **0.99%** | (3.0h) |
| GMS-DH | 0.82 | 1.75% | (3.5s) | 0.86 | 2.96% | (16s) | 0.88 | 3.44% | (1.3m) |
| GMS-DH (aug) | 0.83 | 1.21% | (23s) | 0.86 | 2.38% | (2.2m) | 0.89 | 2.98% | (9.8m) |

Table 11: MGMOCVRP on FLEX$x$ and FIX$x$ with different instances sizes (20, 50 and 100).

| | FLEX2-20 | | | FLEX2-50 | | | FLEX2-100 | | |
| --- | --- | --- | --- | --- | --- | --- | --- | --- | --- |
| | HV | Gap | Time | HV | Gap | Time | HV | Gap | Time |
| HGS | 0.76 | 0.00% | (2.3h) | 0.89 | 0.00% | (15h) | 0.89 | 0.00% | (75h) |
| OR Tools | 0.71 | 6.30% | (1.3h) | 0.84 | 5.93% | (1.5h) | 0.84 | 5.97% | (1.9h) |
| NN | 0.55 | 28.04% | (4.5s) | 0.72 | 18.73% | (18s) | 0.73 | 17.52% | (1.0m) |
| MOGLS | 0.54 | 29.34% | (4.4h) | 0.46 | 48.56% | (49h) | 0.24 | 73.18% | (122h) |
| MOEAD | 0.65 | 14.14% | (5.9h) | 0.68 | 23.83% | (36h) | 0.56 | 37.47% | (218h) |
| NSGA-II | 0.69 | 9.78% | (5.7h) | 0.71 | 20.05% | (37h) | 0.56 | 36.54% | (219h) |
| NSGA-III | 0.68 | 11.02% | (5.8h) | 0.70 | 21.32% | (36h) | 0.55 | 38.08% | (224h) |
| MBM | 0.72 | 5.68% | (5.7s) | 0.83 | 6.07% | (16s) | 0.81 | 9.38% | (54s) |
| MBM (aug) | **0.73** | **4.34%** | (37s) | 0.84 | 5.23% | (2.0m) | 0.82 | 8.30% | (6.9m) |
| GMS-EB | 0.71 | 6.06% | (5.6s) | 0.85 | 4.12% | (1.2m) | 0.85 | 4.08% | (10m) |
| GMS-EB (aug) | 0.73 | 4.36% | (43s) | **0.86** | **3.55%** | (9.0m) | **0.86** | **3.66%** | (1.4h) |
| GMS-DH | 0.70 | 7.83% | (4.3s) | 0.85 | 4.86% | (20s) | 0.85 | 4.94% | (1.0m) |
| GMS-DH (aug) | 0.72 | 4.94% | (24s) | 0.85 | 3.86% | (2.1m) | 0.85 | 3.89% | (8.2m) |
| | FIX2-20 | | | FIX2-50 | | | FIX2-100 | | |
| HGS | 0.77 | 0.00% | (2.5h) | 0.87 | 0.00% | (15h) | 0.91 | 0.00% | (74h) |
| OR Tools | 0.74 | 4.68% | (1.3h) | 0.83 | 5.14% | (1.5h) | 0.88 | 3.78% | (1.9h) |
| NN | 0.61 | 20.78% | (4.5s) | 0.74 | 15.38% | (18s) | 0.81 | 10.74% | (1.0m) |
| MOGLS | 0.54 | 30.14% | (4.4h) | 0.46 | 46.95% | (49h) | 0.36 | 60.78% | (121h) |
| MOEAD | 0.62 | 19.81% | (6.3h) | 0.62 | 28.76% | (36h) | 0.60 | 34.12% | (217h) |
| NSGA-II | 0.66 | 15.14% | (6.1h) | 0.65 | 25.16% | (36h) | 0.61 | 33.28% | (224h) |
| NSGA-III | 0.65 | 16.03% | (6.2h) | 0.64 | 26.86% | (38h) | 0.59 | 35.19% | (220h) |
| MBM | 0.73 | 5.87% | (6.0s) | 0.84 | 4.14% | (16s) | 0.87 | 5.00% | (54s) |
| MBM (aug) | **0.74** | **3.69%** | (38s) | 0.84 | 3.39% | (1.9m) | 0.87 | 4.32% | (6.9m) |
| GMS-EB | 0.74 | 4.67% | (5.5s) | 0.84 | 3.47% | (1.2m) | 0.89 | 2.81% | (9.9m) |
| GMS-EB (aug) | 0.74 | 3.71% | (44s) | **0.85** | **2.71%** | (11m) | **0.89** | **2.38%** | (1.4h) |
| GMS-DH | 0.73 | 5.95% | (4.3s) | 0.83 | 4.96% | (20s) | 0.88 | 3.58% | (1.0m) |
| GMS-DH (aug) | 0.74 | 3.87% | (24s) | 0.84 | 3.96% | (2.1m) | 0.88 | 3.05% | (8.1m) |

Table 12: MGMOTSPTW on FLEX$x$ and FIX$x$ with different instances sizes (20, 50).

| | FLEX2-20 | | | FLEX2-50 | | | FIX2-20 | | | FIX2-50 | | |
| --- | --- | --- | --- | --- | --- | --- | --- | --- | --- | --- | --- | --- |
| | HV | Gap | Time | HV | Gap | Time | HV | Gap | Time | HV | Gap | Time |
| Implicit LKH | 0.27 | 66.74% | (1.1m) | 0.14 | 84.11% | (6.7m) | 0.28 | 67.24% | (1.3m) | 0.15 | 84.24% | (6.5m) |
| Implicit OR Tools | 0.24 | 70.78% | (3.4m) | 0.13 | 85.34% | (25m) | 0.25 | 71.02% | (3.4m) | 0.14 | 85.49% | (26m) |
| MOGLS | 0.64 | 22.92% | (2.7h) | 0.35 | 60.96% | (21h) | 0.62 | 28.85% | (2.8h) | 0.41 | 55.72% | (21h) |
| NSGA-II | 0.69 | 16.55% | (2.4h) | 0.51 | 43.47% | (35h) | 0.73 | 15.93% | (2.7h) | 0.56 | 39.66% | (35h) |
| NSGA-III | 0.69 | 16.01% | (2.5h) | 0.50 | 44.53% | (36h) | 0.71 | 18.88% | (2.5h) | 0.53 | 43.42% | (35h) |
| MOEA/D | 0.75 | 9.19% | (2.7h) | 0.60 | 32.79% | (34h) | 0.74 | 14.99% | (2.8h) | 0.60 | 35.73% | (36h) |
| MBM | 0.72 | 12.89% | (4.9s) | 0.80 | 11.09% | (14s) | 0.53 | 38.64% | (4.9s) | 0.64 | 31.18% | (14s) |
| MBM (aug) | 0.73 | 11.98% | (34s) | 0.81 | 10.10% | (1.8m) | 0.54 | 38.10% | (35s) | 0.66 | 28.58% | (1.9m) |
| GMS-EB | 0.81 | 1.95% | (4.3s) | 0.89 | 0.97% | (60s) | 0.86 | 1.17% | (4.3s) | 0.93 | 0.55% | (59s) |
| GMS-EB (aug) | **0.82** | **0.00%** | (31s) | **0.90** | **0.00%** | (7.9m) | **0.87** | **0.00%** | (31s) | **0.93** | **0.00%** | (7.9m) |
| GMS-DH | 0.80 | 3.37% | (3.3s) | 0.85 | 5.10% | (13s) | 0.85 | 2.47% | (3.4s) | 0.91 | 1.74% | (12s) |
| GMS-DH (aug) | 0.82 | 0.33% | (20s) | 0.87 | 2.69% | (1.6m) | 0.87 | 0.46% | (20s) | 0.92 | 0.78% | (1.7m) |
| GMS-DH Simple | 0.76 | 8.18% | (3.3s) | 0.83 | 7.01% | (13s) | 0.76 | 12.07% | (3.2s) | 0.85 | 8.93% | (12s) |

Table 13: MOTSP for instances with more nodes.

| | EUC150 | | | EUC200 | | |
|---|---|---|---|---|---|---|
| | HV | Gap | Time | HV | Gap | Time |
| LKH | 0.73 | 0.00% | (1.6h) | 0.76 | 0.00% | (3.0h) |
| OR Tools | **0.72** | **1.38%** | (1.7h) | **0.75** | **1.24%** | (3.5h) |
| CNH ZS | 0.72 | 2.04% | (50s) | 0.74 | 2.22% | (1.3m) |
| GMS-EB ZS | 0.71 | 2.71% | (6.7m) | 0.74 | 2.93% | (17m) |
| GMS-DH ZS | 0.71 | 2.82% | (30s) | 0.74 | 3.12% | (1.0m) |
| | TMAT150 | | | TMAT200 | | |
| | HV | Gap | Time | HV | Gap | Time |
| LKH | 0.65 | 0.00% | (39m) | 0.67 | 0.00% | (1.2h) |
| OR Tools | 0.60 | 7.57% | (3.3h) | 0.62 | 7.89% | (6.5h) |
| GMS-EB ZS | **0.64** | **1.64%** | (6.7m) | **0.66** | **1.86%** | (17m) |
| GMS-DH ZS | 0.63 | 3.20% | (30s) | 0.65 | 3.59% | (1.0m) |
| | XASY150 | | | XASY200 | | |
| | HV | Gap | Time | HV | Gap | Time |
| LKH | 0.93 | 0.00% | (40m) | 0.95 | 0.00% | (1.3h) |
| OR Tools | 0.88 | 5.12% | (2.2h) | 0.90 | 4.46% | (4.0h) |
| GMS-EB ZS | **0.90** | **2.77%** | (6.7m) | **0.92** | **2.78%** | (17m) |
| GMS-DH ZS | 0.88 | 5.10% | (30s) | 0.90 | 5.22% | (1.0m) |

Table 14: MGMOTSP for instances with more edges. The GMS models are those trained for FLEX5-100 and FIX5-100.

| | FLEX10-100 | | | FIX10-100 | | |
|---|---|---|---|---|---|---|
| | HV | Gap | Time | HV | Gap | Time |
| LKH | 0.98 | 0.00% | (16m) | 0.86 | 0.00% | (15m) |
| OR Tools | **0.98** | **0.87%** | (57m) | 0.84 | 1.63% | (1.1h) |
| GMS-EB ZS | 0.98 | 0.90% | (23m) | **0.84** | **1.49%** | (23m) |
| GMS-DH ZS | 0.97 | 1.53% | (1.1m) | 0.82 | 4.11% | (1.1m) |

Table 15: MGMOTSP for instances with more nodes.

| | FLEX2-150 | | | FLEX2-200 | | |
|---|---|---|---|---|---|---|
| | HV | Gap | Time | HV | Gap | Time |
| LKH | 0.96 | 0.00% | (39m) | 0.97 | 0.00% | (1.2h) |
| OR Tools | 0.93 | 2.93% | (2.1h) | 0.94 | 2.53% | (3.9h) |
| GMS-EB ZS | **0.94** | **1.78%** | (18m) | **0.95** | **1.74%** | (50m) |
| GMS-DH ZS | 0.94 | 2.17% | (1.0m) | 0.95 | 2.27% | (2.2m) |
| | FLEX5-150 | | | FLEX5-200 | | |
| | HV | Gap | Time | HV | Gap | Time |
| LKH | 0.98 | 0.00% | (39m) | 0.98 | 0.00% | (1.2h) |
| OR Tools | 0.97 | 1.35% | (2.1h) | 0.97 | 1.15% | (4.1h) |
| GMS-EB ZS | **0.97** | **1.05%** | (45m) | **0.97** | **1.02%** | (2.1h) |
| GMS-DH ZS | 0.96 | 1.86% | (1.7m) | 0.96 | 2.03% | (3.5m) |
| | FIX2-150 | | | FIX2-200 | | |
| | HV | Gap | Time | HV | Gap | Time |
| LKH | 0.96 | 0.00% | (39m) | 0.97 | 0.00% | (1.2h) |
| OR Tools | 0.94 | 1.96% | (2.2h) | 0.95 | 1.73% | (4.0h) |
| GMS-EB ZS | **0.95** | **1.35%** | (18m) | **0.96** | **1.34%** | (50m) |
| GMS-DH ZS | 0.94 | 2.38% | (1.1m) | 0.94 | 2.56% | (2.2m) |
| | FIX5-150 | | | FIX5-200 | | |
| | HV | Gap | Time | HV | Gap | Time |
| LKH | 0.92 | 0.00% | (37m) | 0.93 | 0.00% | (1.2h) |
| OR Tools | 0.91 | 1.69% | (2.5h) | 0.92 | 1.59% | (4.5h) |
| GMS-EB ZS | **0.92** | **1.02%** | (46m) | **0.92** | **0.98%** | (2.1h) |
| GMS-DH ZS | 0.89 | 3.99% | (1.7m) | 0.89 | 4.46% | (3.5m) |

Table 16: Tri-objective MOTSP with different instance sizes and distributions. LKH and OR-tools have 105 subproblems, while the number of preferences for the neural methods are in parentheses.

| | EUC20 HV | Gap | Time | EUC50 HV | Gap | Time | EUC100 HV | Gap | Time |
|---|---|---|---|---|---|---|---|---|---|
| LKH | 0.33 | 0.00% | (1.7m) | 0.36 | 0.75% | (14m) | 0.46 | 0.35% | (1.3h) |
| OR Tools | 0.33 | 1.69% | (2.2m) | 0.35 | 4.08% | (15m) | 0.45 | 3.41% | (1.3h) |
| MOGLS | **0.33** | **0.24%** | (1.7h) | **0.36** | **0.00%** | (13h) | **0.46** | **0.00%** | (97h) |
| NSGA-II | 0.28 | 14.99% | (2.3h) | 0.26 | 27.09% | (16h) | 0.32 | 31.75% | (70h) |
| NSGA-III | 0.28 | 16.43% | (2.3h) | 0.25 | 29.36% | (15h) | 0.31 | 33.95% | (66h) |
| MOEA/D | 0.25 | 23.47% | (2.7h) | 0.24 | 34.41% | (16h) | 0.28 | 39.74% | (68h) |
| CNH (105) | 0.32 | 4.69% | (4.3s) | 0.32 | 10.12% | (7.8s) | 0.43 | 7.93% | (31s) |
| CNH (105, aug) | 0.32 | 4.15% | (18s) | 0.33 | 8.95% | (46s) | 0.43 | 7.13% | (2.9m) |
| CNH (1035) | 0.33 | 0.99% | (33s) | 0.35 | 1.61% | (1.1m) | 0.46 | 1.06% | (3.9m) |
| GMS-EB (105) | 0.32 | 4.60% | (2.1s) | 0.32 | 10.04% | (26s) | 0.42 | 8.92% | (3.7m) |
| GMS-EB (105, aug) | 0.32 | 4.15% | (15s) | 0.33 | 8.84% | (3.4m) | 0.43 | 8.12% | (29m) |
| GMS-EB (1035) | 0.33 | 0.99% | (21s) | 0.35 | 2.88% | (4.2m) | 0.45 | 2.48% | (36m) |
| GMS-DH (105) | 0.32 | 4.72% | (1.1s) | 0.33 | 9.81% | (3.9s) | 0.42 | 8.94% | (20s) |
| GMS-DH (105, aug) | 0.32 | 4.03% | (5.5s) | 0.33 | 8.84% | (28s) | 0.42 | 8.32% | (2.7m) |
| GMS-DH (1035) | 0.33 | 1.29% | (11s) | 0.35 | 2.58% | (37s) | 0.45 | 2.48% | (3.2m) |

| | TMAT20 HV | Gap | Time | TMAT50 HV | Gap | Time | TMAT100 HV | Gap | Time |
|---|---|---|---|---|---|---|---|---|---|
| LKH | 0.42 | 0.00% | (60s) | 0.40 | 0.00% | (6.8m) | 0.46 | 0.00% | (32m) |
| OR Tools | 0.40 | 4.24% | (3.3m) | 0.37 | 8.45% | (30m) | 0.41 | 9.93% | (2.7h) |
| MOGLS | 0.36 | 14.59% | (28m) | 0.23 | 43.20% | (3.0h) | 0.21 | 53.58% | (21h) |
| NSGA-II | 0.39 | 6.53% | (47m) | 0.29 | 28.19% | (3.6h) | 0.24 | 46.33% | (14h) |
| NSGA-III | 0.39 | 7.39% | (49m) | 0.29 | 28.68% | (3.7h) | 0.24 | 46.84% | (13h) |
| MOEA/D | 0.37 | 11.23% | (1.1h) | 0.28 | 31.68% | (3.9h) | 0.24 | 47.56% | (13h) |
| MBM (105) | 0.39 | 6.49% | (1.1s) | 0.30 | 24.80% | (3.8s) | 0.29 | 36.31% | (20s) |
| MBM (105, aug) | 0.40 | 3.51% | (4.1s) | 0.32 | 20.06% | (28s) | 0.31 | 32.82% | (2.5m) |
| MBM (1035) | 0.40 | 5.29% | (11s) | 0.31 | 23.06% | (37s) | 0.30 | 35.08% | (3.2m) |
| GMS-EB (105) | 0.41 | 2.77% | (2.2s) | 0.38 | 4.66% | (26s) | 0.44 | 4.42% | (3.7m) |
| GMS-EB (105, aug) | 0.41 | 2.29% | (13s) | 0.39 | 3.84% | (3.4m) | 0.44 | 3.76% | (30m) |
| GMS-EB (1035) | **0.41** | **1.43%** | (21s) | **0.39** | **2.28%** | (4.2m) | **0.45** | **2.24%** | (36m) |
| GMS-DH (105) | 0.40 | 3.84% | (1.2s) | 0.37 | 8.50% | (4.1s) | 0.41 | 10.52% | (20s) |
| GMS-DH (105, aug) | 0.41 | 2.98% | (4.3s) | 0.38 | 6.86% | (29s) | 0.41 | 9.05% | (2.6m) |
| GMS-DH (1035) | 0.41 | 2.58% | (11s) | 0.38 | 6.39% | (37s) | 0.42 | 8.61% | (3.2m) |

| | XASY20 HV | Gap | Time | XASY50 HV | Gap | Time | XASY100 HV | Gap | Time |
|---|---|---|---|---|---|---|---|---|---|
| LKH | 0.58 | 0.00% | (1.2m) | 0.66 | 0.00% | (7.1m) | 0.76 | 0.00% | (34m) |
| OR Tools | 0.53 | 7.94% | (3.2m) | 0.58 | 11.40% | (24m) | 0.69 | 9.96% | (1.9h) |
| MOGLS | 0.45 | 22.50% | (26m) | 0.28 | 56.83% | (3.2h) | 0.26 | 66.38% | (23h) |
| NSGA-II | 0.51 | 11.10% | (47m) | 0.45 | 31.12% | (7.3h) | 0.32 | 58.19% | (14h) |
| NSGA-III | 0.51 | 10.72% | (48m) | 0.46 | 29.65% | (7.2h) | 0.33 | 56.41% | (14h) |
| MOEA/D | 0.50 | 13.97% | (1.0h) | 0.42 | 35.95% | (7.5h) | 0.36 | 52.23% | (14h) |
| MBM (105) | 0.51 | 12.22% | (1.2s) | 0.56 | 15.05% | (3.9s) | 0.68 | 11.27% | (19s) |
| MBM (105, aug) | 0.53 | 8.55% | (5.6s) | 0.58 | 12.15% | (28s) | 0.69 | 9.67% | (2.6m) |
| MBM (1035) | 0.52 | 9.96% | (11s) | 0.58 | 11.60% | (38s) | 0.70 | 8.14% | (3.1m) |
| GMS-EB (105) | 0.54 | 5.96% | (2.2s) | 0.60 | 8.76% | (26s) | 0.71 | 7.35% | (3.7m) |
| GMS-EB (105, aug) | 0.55 | 4.45% | (15s) | 0.61 | 7.63% | (3.4m) | 0.71 | 6.91% | (29m) |
| GMS-EB (1035) | **0.56** | **3.11%** | (21s) | **0.62** | **4.72%** | (4.2m) | **0.73** | **4.15%** | (36m) |
| GMS-DH (105) | 0.53 | 7.40% | (1.1s) | 0.56 | 14.19% | (4.1s) | 0.64 | 16.01% | (20s) |
| GMS-DH (105, aug) | 0.55 | 5.06% | (5.6s) | 0.58 | 12.09% | (29s) | 0.66 | 13.81% | (2.7m) |
| GMS-DH (1035) | 0.55 | 4.97% | (11s) | 0.58 | 11.28% | (38s) | 0.66 | 13.92% | (3.2m) |

Table 17: Tri-objective MGMOTSP with different instance sizes and distributions. LKH and OR-tools have 105 subproblems, while the number of preferences for the neural methods are in parentheses.

| | FLEX2-20 | | | FLEX2-50 | | | FLEX2-100 | | |
|---|---|---|---|---|---|---|---|---|---|
| | HV | Gap | Time | HV | Gap | Time | HV | Gap | Time |
| LKH | 0.71 | 0.00% | (1.1m) | 0.76 | 0.00% | (6.9m) | 0.84 | 0.00% | (32m) |
| OR Tools | 0.68 | 4.87% | (3.0m) | 0.71 | 7.19% | (25m) | 0.78 | 6.21% | (1.9h) |
| MOGLS | 0.55 | 22.53% | (3.5h) | 0.22 | 71.03% | (36h) | 0.15 | 82.42% | (270h) |
| NSGA-II | 0.60 | 16.12% | (1.9h) | 0.40 | 47.38% | (21h) | 0.27 | 67.52% | (380h) |
| NSGA-III | 0.60 | 15.30% | (1.8h) | 0.45 | 40.79% | (20h) | 0.30 | 63.64% | (380h) |
| MOEA/D | 0.60 | 16.13% | (3.2h) | 0.44 | 42.45% | (20h) | 0.34 | 59.85% | (370h) |
| MBM (105) | 0.66 | 7.34% | (4.6s) | 0.69 | 9.28% | (14s) | 0.77 | 7.42% | (47s) |
| MBM (105, aug) | 0.69 | 3.04% | (33s) | 0.71 | 6.99% | (1.8m) | 0.78 | 6.17% | (6.1m) |
| MBM (1035) | 0.67 | 5.76% | (46s) | 0.71 | 6.18% | (2.3m) | 0.80 | 4.42% | (7.7m) |
| GMS-EB (105) | 0.67 | 5.30% | (3.9s) | 0.71 | 6.88% | (58s) | 0.79 | 5.46% | (9.4m) |
| GMS-EB (105, aug) | 0.68 | 3.69% | (29s) | 0.72 | 5.68% | (7.8m) | 0.80 | 4.83% | (1.3h) |
| GMS-EB (1035) | 0.69 | 2.66% | (38s) | **0.74** | **3.18%** | (9.5m) | **0.81** | **2.80%** | (1.5h) |
| GMS-DH (105) | 0.68 | 4.10% | (3.1s) | 0.71 | 6.52% | (13s) | 0.79 | 6.07% | (53s) |
| GMS-DH (105, aug) | 0.69 | 2.79% | (18s) | 0.72 | 5.15% | (1.5m) | 0.79 | 5.58% | (6.9m) |
| GMS-DH (1035) | **0.70** | **1.63%** | (30s) | 0.73 | 3.39% | (2.0m) | 0.81 | 3.69% | (8.4m) |

| | FIX2-20 | | | FIX2-50 | | | FIX2-100 | | |
|---|---|---|---|---|---|---|---|---|---|
| LKH | 0.76 | 0.00% | (1.2m) | 0.86 | 0.00% | (7.1m) | 0.90 | 0.00% | (32m) |
| OR Tools | 0.74 | 3.54% | (3.1m) | 0.83 | 3.93% | (26m) | 0.87 | 3.42% | (2.0h) |
| MOGLS | 0.63 | 17.63% | (4.0h) | 0.45 | 47.61% | (36h) | 0.38 | 58.28% | (250h) |
| NSGA-II | 0.63 | 17.65% | (2.2h) | 0.53 | 38.86% | (23h) | 0.44 | 51.33% | (420h) |
| NSGA-III | 0.65 | 15.29% | (2.1h) | 0.54 | 36.93% | (22h) | 0.43 | 52.54% | (460h) |
| MOEA/D | 0.64 | 16.14% | (3.7h) | 0.54 | 36.85% | (19h) | 0.45 | 50.23% | (450h) |
| MBM (105) | 0.73 | 4.04% | (4.6s) | 0.81 | 5.32% | (14s) | 0.87 | 4.13% | (47s) |
| MBM (105, aug) | 0.75 | 1.47% | (33s) | 0.83 | 3.91% | (1.8m) | 0.87 | 3.43% | (6.2m) |
| MBM (1035) | 0.74 | 2.56% | (46s) | 0.83 | 3.86% | (2.3m) | 0.88 | 2.77% | (7.7m) |
| GMS-EB (105) | 0.74 | 3.77% | (3.9s) | 0.83 | 3.64% | (58s) | 0.87 | 3.19% | (9.4m) |
| GMS-EB (105, aug) | 0.74 | 2.73% | (29s) | 0.83 | 3.11% | (7.7m) | 0.88 | 2.89% | (1.3h) |
| GMS-EB (1035) | 0.75 | 1.81% | (38s) | **0.84** | **1.92%** | (9.5m) | **0.89** | **1.76%** | (1.5h) |
| GMS-DH (105) | 0.74 | 3.14% | (3.1s) | 0.83 | 3.88% | (12s) | 0.87 | 3.73% | (54s) |
| GMS-DH (105, aug) | 0.75 | 2.21% | (18s) | 0.83 | 3.52% | (1.5m) | 0.87 | 3.56% | (6.9m) |
| GMS-DH (1035) | **0.76** | **1.24%** | (29s) | 0.84 | 2.22% | (2.0m) | 0.88 | 2.30% | (8.4m) |

