# OpenReview forum: "Beyond Simple Graphs: Neural Multi-Objective Routing on Multigraphs"
_ICLR.cc/2026/Conference — ICLR 2026 Poster_

### Official Review · Reviewer_YhQ4 · 2025-10-27

**Soundness:** 3
**Presentation:** 4
**Contribution:** 3
**Rating:** 8
**Confidence:** 3

**Summary:**

The paper introduces GNN-based Multigraph Solvers (GMS)—two novel neural architectures for multi-objective (MO) routing on multigraphs, a setting previously unaddressed in neural combinatorial optimization. The first variant (GMS-EB) operates directly on the multigraph using edge-based autoregressive decoding, while the second (GMS-DH) prunes the multigraph via a learned edge-selection mechanism before performing node-based decoding. Both models employ preference-conditioned policies for MO optimization and are trained with a REINFORCE-style algorithm adapted for Chebyshev scalarization. Extensive experiments across multiple problem types (MOTSP, MOCVRP, and MGMOTSP) and graph distributions show that GMS achieves competitive or superior performance compared to strong neural and heuristic baselines, including LKH, MOEA/D, and a MatNet-based model.

**Strengths:**

**Originality**

- This is the first neural approach to handle multi-objective routing on multigraphs, an important and realistic extension beyond simple graphs. The paper clearly articulates why existing models (e.g., transformer-based TSP solvers) fail in this setting.

- The dual-decoder architecture (GMS-DH) is elegant, combining non-autoregressive edge pruning and autoregressive routing / node selection in a unified framework. The preference-conditioned hypernetwork design for multi-objective learning is well-justified and grounded in prior work (Lin et al., 2022; Navon et al., 2021).


**Quality & Clarity**
- The paper does an excellent job explaining why existing methods fail—namely, that standard encoders cannot represent multigraphs and node-based decoders are insufficient when edge choice is also required. The connection between multi-objective problems and the need for multigraphs is natural and well-argued.
- clear writing and figures describing the details of the neural policy
- The inclusion of training algorithms, detailed appendices, and reproducibility statement (code included) is commendable.


**Significance**
- Strong Empirical Validation: The experiments are comprehensive and well-designed.

- Strong Baselines: The comparison against state-of-the-art heuristics like LKH3 and a purpose-built neural baseline (MBM)  provides a robust benchmark.

- Crucial Ablation (GMS-DH PP): The "GMS-DH PP" variant, which uses a simple pre-pruning heuristic instead of the learned selection head, performs very poorly (Table 1). This convincingly demonstrates that the learned pruning mechanism of GMS-DH is far superior to a naive adaptation.

**Weaknesses:**

Both proposed variants face notable scalability issues. GMS-DH performs poorly on larger instances with many edges, suggesting that its hierarchical design does not scale effectively. Meanwhile, GMS-EB is computationally impractical for large graphs due to its excessive training and inference time, scaling poorly with the number of edges. Notably, the paper lacks a discussion or outlook on these scalability limitations and provides no guidance or future directions for addressing them.

**Questions:**

Why does performance for GMS-DH degrade on distributions with more edges (FLEX10, FIX10)? How do you think this can be mitigated in future work (without the excessive runtimes of GMS-EB)?

---

> ### Author Response · Authors · 2025-11-20
>
> We thank reviewer YhQ4 for the time they have taken to write the review and for their helpful comments. We are also glad they found the paper fitting for acceptance.
>
> > Both proposed variants face notable scalability issues...
>
> We acknowledge the difficulty of scaling to very large instances with high node counts and many edges. Much of the difficulty inherently comes from the multigraph formulation, which requires some degree of edge manipulation in the model, and the number of edges scales quadratically with the node count. We have added a small discussion along these lines in Appendix H.4 and shortly about future steps in the final section of the manuscript. The new changes are marked in blue.
>
> > Why does performance for GMS-DH degrade on distributions with more edges...
>
> We would argue that GMS-DH still maintains a relatively stable performance even in the case of FLEX10-100 and FIX10-100, although it performs worse than OR-tools and GMS-EB. As for the reason of the slightly worse performance compared to e.g., FLEX2-100 and FIX2-100, we note that both FLEX10-100 and FIX10-100 have many edges that need to be chosen between in the pruning step, which inevitably makes the problem harder as the action space is larger for the pruning decoder. In future work, we will explore alternative decomposition approaches to pruning, that potentially may improve performance.

---

### Official Review · Reviewer_H4sL · 2025-10-30

**Soundness:** 2
**Presentation:** 3
**Contribution:** 3
**Rating:** 4
**Confidence:** 4

**Summary:**

The paper presents GNN-based Multigraph Solver (GMS), the first neural architecture designed for multigraph and capable of handling multi objective routing problems on asymmetric graphs. Specifically, the paper considers two variants: one that is edge-based and one that is node-based and follows an edge pruning stage that converts the multigraph into a simple graph. Experimental results show the approach outperform the baselines for multi-objective TSP and multi objective CVRP problems, particularly on asymmetric graphs and multi graphs.

**Strengths:**

**Strengths:**
- The paper focuses on a relatively unexplored setting: multi-graph multi-objective neural routing optimization
- The proposed dual-head GMS seems novel
- Experiments show clear gains over the baselines

**Weaknesses:**

**Weaknesses:**
- The paper claims that existing techniques are not suitable for asymmetric and multigraph but this claim is not sufficiently substantiated. While approaches that rely solely on selecting the order of nodes are not sufficient, several recent approaches focus instead on selecting edges instead (for example, DIFUSCO and GREAT, both are cited in the paper). These approaches, at least in principle, can be used on asymmetric/multigraphs, and the paper does not compare to them, or explain why they are not suitable. There is no clear explanation why using them in a multigraph is not trivial, or what is the different between such approaches and the proposed edge-based approach (GMS-EB).

- The technical novelty needs to be more clearly specified in the paper: while it is clear that the combined setting of multigraph and multi objective has not been explored, the two are not intertwined in the proposed technical approach (that involves GMS, as well as a preference-condition MLP, and multi objective RL training) and the paper would benefit from making clear claims on the what is novel in each of the components, as well as the combined setting.

- I did not fully understand the claim on the scaling of edge-based GMS: why does it scale as O(MN^4)? If the auto-regressive decoder selects the next edge in the tour then it sounds like there are only MN edges to select from at each step and not MN^2?

- Missing multi-objective neural baselines: there are various works on multi-objective TSP that should be considered as baselines in the MO experiments; following is a subset of recent approaches. Most are not mentioned and all are not compared to:
	* Lin et al., 2022 that is mentioned in the paper is not being used.
	* Wu, Y., Fan, M., Cao, Z., Gao, R., Hou, Y., & Sartoretti, G. (2024, May). Collaborative deep reinforcement learning for solving multi-objective vehicle routing problems. In 23rd International Conference on Autonomous Agents and Multiagent Systems, AAMAS 2024 (pp. 1956-1965). International Foundation for Autonomous Agents and Multiagent Systems (IFAAMAS).
	* Li, S., Wang, F., He, Q., & Wang, X. (2023). Deep reinforcement learning for multi-objective combinatorial optimization: A case study on multi-objective traveling salesman problem. Swarm and Evolutionary Computation, 83, 101398.
	* Wu, R., Wang, R., Hao, J., Wu, Q., Wang, P., & Niyato, D. (2024). Multiobjective vehicle routing optimization with time windows: A hybrid approach using deep reinforcement learning and nsga-ii. IEEE Transactions on Intelligent Transportation Systems.
	* Fan, M., Zhou, J., Zhang, Y., Wu, Y., Chen, J., & Sartoretti, G. A. (2025). Preference-Driven Multi-Objective Combinatorial Optimization with Conditional Computation. arXiv preprint arXiv:2506.08898.

- Not clear why Table 2 and Table 3 do not include all baselines

Minor: please fix the backward opening quote marks in all quotations in the paper.

**Questions:**

See my concerns and questions above

---

> ### Author Response · Authors · 2025-11-20
>
> We thank reviewer H4sL for the time they have spent on reviewing our submission and for their insightful comments. Our responses are as follows. We have edited the manuscript accordingly (changes marked in blue).
>
> >The paper claims that existing techniques are not suitable for asymmetric and multigraphs...
>
> We agree that we can be more explicit in the manuscript with reviewing current techniques and explaining why they fall short. For this purpose we added a new section in Appendix B that addresses this. In short, GREAT, DIFUSCO and some other existing methods can encode multigraphs, but that does not mean that they are directly applicable to routing on multigraphs. Particularly the decoding stage must be modified. Hence one of our main arguments is that current decoding strategies are insufficient. While one can argue that it is in principle possible to extend these models to the multigraph framework, doing so in practice is not trivial. In fact, GMS-EB can be viewed as a rather straight-forward modification to GREAT to handle multigraphs, but as we see its runtime scales quite badly. To our knowledge, our work is the first explicit investigation of neural solvers for multigraph problems.
>
> >The technical novelty needs to be more clearly specified...
>
> We agree that the technical novelty can be clarified further, and we have added Appendix B.3 for this purpose as well as a short remark in Section 4.2.
>
> In short, the novelty in the multi-objective setting is that we are among the first to tackle multi-objective problems on asymmetric graphs, which are much more difficult than one would initially think (as shown by the failure of the "naive" MBM to perform well). We present two models that work well experimentally.
>
> In the multigraph setting, besides being the first and only to explicitly tackle this setting, GMS-EB is rather novel in its edge-selection procedure with a tailored edge-based MP decoder. The combined Non-Autoregressive (NAR) and Autoregressive (AR) nature of GMS-DH is also novel in the edge-pruning + node permutation context, and specifically designed to avoid much of the scaling issues inherent with multigraphs. Additionally, the collaborative RL-based training of the two heads (outlined in Section 4.3) is different compared to other dual-decoder works (e.g., [1]) and NAR + AR work (e.g., [2]). It ensures all parts of the model can be trained simultaneously without high-quality labels.
>
> >I did not fully understand the claim on the scaling...
>
> Yes, in our implementation we only calculate a score for outgoing edges in each node. However, as all nodes will be visited eventually we still need to store and manipulate embeddings for all $MN^2$ edges compared to $N$ nodes in a node-based decoder. This has now been clarified in the paper.
>
> >Missing multi-objective neural baselines...
>
> The papers [3, 4, 5, 6, 7] exclusively tackle Euclidean problems (they are transformer-based working with node coordinates and hence not equipped to encode asymmetry). The exception is POCCO [5], a model-agnostic framework, but it is only evaluated with transformer-based models on Euclidean instances and integrating POCCO with GMS is beyond the scope of this work. Euclidean problems are not our focus in this paper. Thus we only briefly report performance on Euclidean instances in the appendix compared to one recent work (CNH). However, we have ensured that we cite all the listed papers in the literature review in Appendix A.2.
>
> >... Table 2 and Table 3 do not include all baselines.
>
> Table 3 contains all relevant baselines (LKH and OR-tools are not well-defined on this problem).
>
> In Table 2 we are missing MBM and the MOEAs. The latter are far from competitive for 100 nodes, so including them would serve little purpose for higher node counts. The former is not included because its input size is limited by its embedding size (128, see https://openreview.net/forum?id=z2z9suDRjw for a discussion). We clarified these points further in the paper.
>
> References
>
> [1] “Learning to Handle Complex Constraints for Vehicle Routing Problems.” Neurips. 2024
>
> [2] “GLOP: Learning Global Partition and Local Construction for Solving Large-scale Routing Problems in Real-time,” AAAI. 2024
>
> [3] “Pareto Set Learning for Neural Multi-objective Combinatorial Optimization.” ICLR. 2022
>
> [4] “Collaborative deep reinforcement learning for solving multi-objective vehicle routing problems.” International Conference on Autonomous Agents and Multiagent Systems. 2024
>
> [5] “Deep reinforcement learning for multi-objective combinatorial optimization: A case study on multi-objective traveling salesman problem.” Swarm and Evolutionary Computation. 2023
>
> [6] “Multiobjective vehicle routing optimization with time windows: A hybrid approach using deep reinforcement learning and nsga-ii.” IEEE Transactions on Intelligent Transportation Systems. 2024
>
> [7] “Preference-Driven Multi-Objective Combinatorial Optimization with Conditional Computation.” Arxiv. 2025

---

### Official Review · Reviewer_6Fgk · 2025-10-31

**Soundness:** 3
**Presentation:** 4
**Contribution:** 3
**Rating:** 4
**Confidence:** 3

**Summary:**

This work represents an extension of some NCO models to multi-objective optimization on multigraphs. In particular, the authors propose a novel method for selecting edges from the multigraph, rather than only nodes, which is typically the case in the neural combinatorial optimization (NCO) domain.

Two different autoregressive approaches are presented: the first operates directly on multigraphs, while the second performs pruning on the multigraph before executing routing on the simplified graph. The proposed method is evaluated on several benchmarks, including the multi-objective (MO) TSP, MO-CVRP, multigraph (MG) MO-TSP, and MG-MO-TSP with time windows.

**Strengths:**

1. The paper is well organized and written in a clear manner.
2. The proposed incorporation of edge selection into the decoding process represents a potentially novel contribution to the NCO domain.
3. The inclusion of two complementary models, one prioritizing speed and the other performance, is a reasonable design choice. However, the experimental results do not provide sufficient justification for maintaining both variants.

**Weaknesses:**

1. Contribution claims are overstated. The authors state that existing methods rely on transformers and can handle only problems defined on simple graphs, not beyond routing problems in the Euclidean settings. However, there are several works capable of encoding complex graphs, and the authors even cite some of them, although they claim that, to the best of their knowledge, such methods do not exist. Two of the cited works, (Kwon et al., 2021) and (Drakulic et al., 2025), can encode multigraphs and MatNet is even used in this work.

2. Related to the above, the novelty is limited. The main contribution lies in the implementation of the idea of autoregressively selecting edges from the hypergraph.

3. Limited evaluation. Although this work focuses on routing on multigraphs - which is its main contribution - it is evaluated on only one problem type (with two variants) of MGMOTSP. As the authors state, there is no prior work or benchmark datasets for this type of problem, which makes the list of baselines very limited. In this setting, it is difficult to assess the quality of the proposed solution. Relatedly, it is unclear what the broader impact of this work would be on the NCO community, which so far has not shown interest in studying these types of problems.

**Questions:**

1. I do not fully understand the experimental setup for testing MOTSP and MOCVRP with the Edge-based GMS. It seems that, during decoding, these problems do not require selecting edges from the hypergraph, only nodes. Could you please clarify how exactly this model is applied?

2. Is it possible to solve MOTSP and MOCVRP using classical NCO solvers such as the “vanilla” GREAT, MatNet, or GOAL models, adapted to the multi-objective setting? If so, I would like to see them included in Table 1 as baselines.

3. You provide a neural baseline based on the MatNet architecture, which is very nice, and demonstrate that your method is plug-and-play and compatible with other architectures. I would, however, like to see the results of the same experiments built upon other existing models that handle multigraphs.

4. Your experiments show that the edge-based model is only slightly better than the second model, but much slower, which raises questions about its purpose. It is approximately 10× slower while providing at most a 1% improvement in terms of the gap, which is quite negligible. Could it be that the selected benchmarks are too easy for both models, and that this is why the first model cannot demonstrate its full potential? Could you generate some more challenging problem instances and run tests of them?

5. The same question as above applies to GMS-DH PP — what is its purpose? For a small improvement in running time, the performance drops significantly.

---

> ### Author Response · Authors · 2025-11-20
>
> We thank reviewer 6Fgk for the helpful comments. We address all concerns below and updated the manuscript accordingly (changes in blue). New experiments are pending; placeholder tables have been added and all results will be completed before the deadline.
>
> > W1
>
> Many prior works [1, 2, 3, 4, 5] can encode asymmetric graphs. However, most of these, in particular transformer-based architectures like MatNet [1] and GOAL [2], are unsuitable for encoding multigraphs. They rely on distance matrices as input to mixed attention blocks, and representing multigraphs as $N\times N\times D$ tensors (e.g., via concatenation) breaks permutation invariance across parallel edges. Note, we use MatNet on multigraphs by pre-pruning so that there is only one edge between each node pair. This works, but learned pruning after encoding (like in GMS-DH) is more suitable.
>
> Some methods [3, 5] (GNN-based) can encode multigraphs, but to our knowledge none of them have been applied in this setting. Then, they would also need to be accompanied with some additional decoding strategy. To clarify further, we added Appendix B.1 with discussion about inadequacies of current methods and changed wording in the introduction.
>
> > W2
>
> The novelty goes beyond autoregressive edge-selection. It lies mainly in the decoding (edge-based for GMS-EB and two-stage pruning + routing for GMS-DH), including associated training algorithms + architectures, as well as overall framework and analysis of learning-for-routing in the multigraph context. To clarify the technical novelty further, we added Appendix B.3 and a short remark in Section 4.2.
>
> > W3a. Limited evaluation...
>
> As the reviewer points out, there are few baselines for these problems. This is why we emphasize MGMOTSP, where LKH provides a near-optimal baseline, making it easy to assess the performance of our models. However, we agree that investigating a different problem apart from MGMOTSP and MGMOTSPTW is a good idea. We added space in Table 1 for results for a multigraph MOCVRP (with HGS as near-optimal baseline) and Appendix E.5 to explain the new problem.
>
> > W3b. Relatedly, it is unclear what the broader impact of this work would be...
>
> In recent years, the NCO community has been moving towards more challenging settings, e.g., dynamic problems [6], constrained problems [7] and problems in real world networks [8]. These settings are very interesting to the OR community and hold inherent challenges for the NCO community. Our work can be seen as a part of a transition from “simple” VRPs, e.g., the Euclidean TSP, towards more intricate settings. We hope that the paper will inspire more works tackling multigraph problems, which offer rich structure and scaling challenges.
>
> > Q1 & Q2
>
> Correct, multi-objective problems on simple graphs do not require edge selection and it is possible to solve these using “naive” extensions of existing methods for single-objective problems. However, we find that simple extensions using hypernetworks in the decoder underperform compared to our more intricate GMS. In the original submission we included the natural extension of MatNet (the MBM benchmark). Now we added similar results for GREAT and GOAL in Appendix H.1 (table to be filled in). Thank you for this suggestion. We also clarified in the introduction that extending current methods is easy but yields subpar performance.
>
> > Q3
>
> As with Q2, we added these results in Appendix H.1.
>
> > Q4
>
> Yes, in most cases GMS-EB only slightly outperforms GMS-DH, but there are exceptions. See e.g., Table 3, where it outperforms by 2.69%. Also note XASY200 in Table 2 and the tri-objective results in Table 14. In Figures 5 and 6 it can also be seen that GMS-EB is more sample-efficient than GMS-DH. Hence, we think it is warranted to include GMS-EB.
>
> Regarding more challenging problems, the MGMOTSPTW in Table 3 is quite challenging, with baselines severely underperforming compared to GMS. We also think that the new MGMOCVRP constitutes a more difficult problem.
>
> > Q5
>
> GMS-DH PP is an ablation of the pruning decoder, demonstrating that learned pruning gives faster inference and better performance than pre-pruning. This is now clarified.
>
> References
>
> [1] Y.-D. Kwon et al., “Matrix Encoding Networks for Neural Combinatorial Optimization.” Neurips. 2021.
>
> [2] D. Drakulic et al., “GOAL: A Generalist Combinatorial Optimization Agent Learning.” ICLR, 2025.
>
> [3] “A GREAT Architecture for Edge-Based Graph Problems Like TSP.” Arxiv. 2024.
>
> [4] “UniteFormer: Unifying Node and Edge Modalities in Transformers for Vehicle Routing Problems.” Neurips, 2025.
>
> [5] “DIFUSCO: Graph-based Diffusion Solvers for Combinatorial Optimization.” Neurips. 2023.
>
> [6] “Neural Combinatorial Optimization for Time Dependent Traveling Salesman Problem.” Neurips. 2025.
>
> [7] “Learning to Handle Complex Constraints for Vehicle Routing Problems.” Neurips. 2024.
>
> [8] “Neural Combinatorial Optimization for Real-World Routing.” Arxiv. 2025.

---

> > ### Comment · Reviewer_6Fgk · 2025-11-21
> >
> > Thank you very much for your answers and clarifications.
> >
> > You are right about the issue of breaking permutation invariance across parallel edges in transformer-based approaches, which I completely overlooked in my first reading. I still feel that existing approaches could be extended to a similar setting, but that is beyond the scope of your paper.
> >
> > I am glad to hear that you added more baselines for MOTSP and MOCVRP and that you claim to outperform them. However, it seems that Table 6 is broken - it does not include the new baselines, and it currently has no content at all.
> >
> > Please fix this, and I will increase my score.

---

> > > ### Author Response · Authors · 2025-11-21
> > >
> > > Thank you for taking the time to revisit the manuscript.
> > >
> > > Regarding Table 6, many of the new experiments are still running, which is why the table currently appears empty. We wanted to ensure that all methods are evaluated consistently before finalizing the results. As soon as all runs are completed, we will update Table 6. We apologize for the confusion
> > >
> > > We will post a comment to notify the reviewers once the final manuscript with all completed tables is uploaded.
> > >
> > > Thank you again for your constructive comments and for considering increasing your score once the table is fixed.

---

> > > ### Author Response · Authors · 2025-11-27
> > >
> > > We have now updated the submission with all experiments as promised. This includes the new MGMOCVRP results (see the bottom of Table 1 and Table 11 in the appendix) as well as the newly added baselines in Table 6.
> > >
> > > Thank you very much for your continued effort in the review process.

---

> ### Comment · Reviewer_6Fgk · 2025-11-28
>
> Thank you for the updates.
>
> I am glad to see that your approach performs (surprisingly to me) much better than existing models on MO problems in node-based construction.
>
> In my opinion, this makes your work much stronger, opens the door to additional applications, and clearly deserves acceptance.
>
> I am not sure what happened, I would like to increase my score to 8 - accept, but I can not edit my review anymore. As I mentioned, I support the acceptance, and I will increase the score if it becomes possible.

---

> ### Author Response · Authors · 2025-11-28
>
> Thank you for further raising your score and for supporting the acceptance of the paper.
>
> We will release the code and results publicly once the paper is accepted, supporting open science and the full reproducibility of our results.

---

### Comment · Area_Chair_d8dD · 2025-11-24
**Discussion Period**

Dear reviewers,

The discussion period is now open. Please use the “Official Comments” to engage in discussions about each other's reviews and the authors' rebuttal, and update your assessments or comments as appropriate.

Did the authors' rebuttal adequately address your concerns? We kindly ask that you update your reviews based on these discussions and your evaluation of the rebuttal, even if your overall assessment remains unchanged.

Thank you all for your contributions.

Best regards, AC

---

### Author Response · Authors · 2025-11-27

Dear reviewers,

We would like to inform you that we have now completed all planned experiments and updated the manuscript accordingly. All revisions relative to the original submission are highlighted in blue.

Thank you again for your helpful feedback and for your continued efforts throughout the review process.

---

### Author Response · Authors · 2025-12-01
**Discussion Period Summary**

Dear area chair.

Due to the abrupt end to the discussion phase, we would like to summarize the discussions as follows:
- **Reviewer YhQ4 (score 8)** maintains their score. We have answered the few remarks they had below, and by editing the manuscript.
- **Reviewer 6Fgk (initial score 4, raised to 8)** now strongly supports the acceptance of the paper. We addressed all of their concerns, including adding new experiments and clarifying the novelty in the manuscript.
- **Reviewer H4sL (score 4)** did not respond to our rebuttal. We would like to note the following regarding their concerns 1-5:
  - We believe **concerns 3-5** have been addressed thoroughly by minor remarks in the article. As a whole, these points appear to stem from misunderstandings, which we clarified in the revised manuscript.
  - **Concerns 1-2** are regarding novelty. To further clarify, we have added two sections in the Appendix and some smaller edits to the main text.
  - Regarding the novelty, we also remark that both reviewers YhQ4 and 6Fgk appreciate the originality and significance of the work.
- Consequently, had the discussion phase continued, it seems likely that **reviewer H4sL** might have increased their score.

Thank you for potentially taking these comments into account in the final recommendation.

---

### Meta-Review · Area_Chair_mjqC · 2025-12-30

**Summary:**

This work proposes two GNN-based Multigraph Solvers (GMS) for tackling multi-objective routing problems on multigraphs, namely Edge-based GMS (GMS-EB) and Dual Head GMS (GMS-DH). GMS-EB directly constructs tours by autoregressively selecting edges on the multigraph, while GMS-DH first prunes the multigraph through a learned selection step before performing autoregressive routing on the pruned graph. Both models demonstrate competitive performance across various multi-objective multigraph routing problems.

The reviewers initially had mixed scores (8,4,4) and raised concerns regarding the claimed contributions, novelty, scalability, evaluation, clarity, experimental settings, and performance. Most concerns have been properly addressed by the rebuttal. After rebuttal, Reviewer 6Fgk raised their score from 4 to 8, and I believe the major concerns from Reviewer H4sL have also been adequately resolved. As a result, the final scores could be (8, 8, 6).

I read this paper in detail and agree with the reviewers that this work is well-written and easy to follow, the multi-objective routing problem on multigraphs is important yet underexplored, the proposed method is novel and reasonable, and the experimental results are promising. Therefore, I recommend accepting this work.

**Reviewer Concerns:**

I believe most concerns have been properly addressed by the rebuttal, and no major issues remain that would prevent the acceptance of this work.

**Reviewer Scores:**

If the reviewers had been able to participate fully in the discussion, I am confident that Reviewer 6Fgk would have raised their score from 4 to 8, as reflected in the author-reviewer discussion. Meanwhile, I believe Reviewer H4sL is likely to adjust their score from 4 to 6, given that at least their main concerns have been properly addressed in the rebuttal. Therefore, the final scores for this submission could reasonably be (8, 8, 6).

---

### Decision · Program_Chairs · 2026-01-26

Accept (Poster)